# A new class of type VI secretion system effectors can carry two toxic domains and are recognized through the WHIX motif for export

**Chaya Mushka Fridman[1], Kinga Keppel[1], Vladislav Rudenko[1], Jon Altuna-Alvarez[2], David Albesa-Jové[2,3,4], Eran Bosis**[5]\*, **Dor Salomon**[1]\*

1 Department of Clinical Microbiology and Immunology, School of Medicine, Faculty of Medical and Health Sciences, Tel Aviv University, Tel Aviv, Israel, 2 Instituto Biofisika (CSIC, UPV/EHU), Fundación Biofísica Bizkaia/Biofisika Bizkaia Fundazioa (FBB), Leioa, Spain, 3 Departamento de Bioquímica y Biología Molecular, University of the Basque Country, Leioa, Spain, 4 Ikerbasque, Basque Foundation for Science, Bilbao, Spain, 5 Department of Biotechnology Engineering, Braude College of Engineering, Karmiel, Israel

\* bosis@braude.ac.il (EB); dorsalomon@mail.tau.ac.il (DS)

## Abstract

Gram-negative bacteria employ the type VI secretion system (T6SS) to deliver toxic effectors into neighboring cells and outcompete rivals. Although many effectors have been identified, their secretion mechanism often remains unknown. Here, we describe WHIX, a domain sufficient to mediate the secretion of effectors via the T6SS. Remarkably, we find WHIX in T6SS effectors that contain a single toxic domain, as well as in effectors that contain two distinct toxic domains fused to either side of WHIX. We demonstrate that the latter, which we name double-blade effectors, require two cognate immunity proteins to antagonize their toxicity. Furthermore, we show that WHIX can be used as a chassis for T6SS-mediated secretion of multiple domains. Our findings reveal a new class of polymorphic T6SS cargo effectors with a unique secretion domain that can deploy two toxic domains in one shot, possibly reducing recipients' ability to defend themselves.

## Introduction

Interbacterial competition can be a strong driver of bacterial evolution [1,2]. A major player in bacterial warfare is the type VI secretion system (T6SS), an offensive apparatus widespread in Gram-negative bacteria [3–7]. The T6SS delivers a cocktail of antibacterial toxins, called effectors, into neighboring bacteria in a contact-dependent manner [5,8–12]. Notably, some T6SSs deliver effectors that target eukaryotic cells and play a role in virulence or protection against predation [7,13–21]. Upon T6SS activation, a contractile sheath propels a tube that is made of hexameric Hcp rings and capped with a spike comprising VgrG and PAAR or PAAR-like repeat-containing proteins (the latter will be collectively referred to as PAAR hereafter) out of the cell; the tube-spike, which is decorated with effectors, then penetrates a neighboring bacterium where the effectors are deployed [22].

**Data availability statement:** The mass spectrometry proteomics data have been deposited in the ProteomeXchange Consortium via the PRIDE partner repository with the dataset identifier PXD054809. The data can be downloaded via https://ftp.pride.ebi.ac.uk/pride/data/archive/2024/08/PXD054809. The whole genome sequence of *A. jandaei* DSM 7311 is available as NCBI RefSeq assembly GCF_037890695.1; Chromosome RefSeq accession NZ_CP149571.1. The AlphaFold-produced structure models used in this work are available as Supporting information files.

**Funding:** This project received funding from the Israel Science Foundation (ISF grant number 1362/21 to DS and EB). D.A.-J. acknowledges support from PID2021-127816NB-I00/AEI/10.13039/501100011033/FEDER, UE; and IT1745-22/Basque Government. CMF was supported by a scholarship from the Clore Israel Foundation and by a scholarship for outstanding doctoral students from the Orthodox community from the Council for Higher Education. The funders played no role in the study design, data collection and analysis, decision to publish, or preparation of the manuscript.

**Competing interests:** The authors have declared that no competing interests exist.

**Abbreviations :** Awe1, *Aeromonas* <u>W</u>HIX effector <u>1</u>; CFU, colony forming units; GFP, green fluorescent protein; GST, glutathione S-transferase; LB, lysogeny broth; MCS, multiple cloning site; PAE, Predicted Aligned Error; PSSM, position-specific scoring matrix; T6SS, type VI secretion system; WHIX, <u>WH</u> type s<u>IX</u> motif.

Many antibacterial T6SS effector families have been investigated to date, revealing toxic mechanisms in the bacterial cytoplasm, periplasm, and membrane, including peptidoglycan-degrading enzymes [8,23,24], pore-forming toxins [25–27], phospholipases [10,28], nucleases [29–32], NAD(P)$^+$-degrading enzymes [33,34], effectors inhibiting protein synthesis [35] or cell division [36], and ADP-ribosyl transferases [36,37]. Importantly, antibacterial effectors are encoded next to a cognate immunity protein that prevents self/kin-intoxication [5,38–40].

T6SS effector repertoires include "specialized effectors"—tube-spike components fused to a toxic domain, and "cargo effectors"—toxic domain-containing proteins that non-covalently bind a secreted tube-spike component [41,42], often via a "loading platform" region at its C-terminus [28,43,44]. Some cargo effectors require an adaptor protein [45,46], a secreted adaptor, or a co-effector for proper loading and secretion [47,48].

Cargo effectors that load onto the spike tend to be modular proteins comprising a C-terminal toxic domain and an N-terminal secretion domain responsible for loading the effector onto the spike "loading platform" [30,48–51]. We previously identified several non-structural T6SS-specific secretion domains named MIX [49], RIX [48], and PIX [51], each defining a class of polymorphic T6SS effectors. We showed that these domains are necessary and sufficient to mediate secretion through the T6SS. An additional domain, FIX, was also used to define a widespread class of polymorphic T6SS effectors, yet its role in T6SS secretion remains unclear [32]. Although many T6SS effectors have been described, we hypothesize that additional T6SS secretion domains, which can be used to reveal new classes of polymorphic T6SS effectors and secretion mechanisms, are yet to be identified.

Here, we describe a T6SS secretion domain named WHIX. We reveal that WHIX domains define a widespread class of polymorphic effectors containing predominantly periplasm-targeting toxic domains. Remarkably, we demonstrate that WHIX domains, which are sufficient to mediate T6SS-depended secretion, can be fused to either one or two toxic domains, each requiring its own cognate immunity protein for protection.

## Results

### WHIX is a widespread T6SS-specific domain found in peptidoglycan-targeting effectors

In previous work, we determined the effector repertoires of T6SS1 in *Vibrio alginolyticus* 12G01 [52] and *Vibrio coralliilyticus* BAA-450 [21]. While examining these effector repertoires, we observed that two proteins, one from *V. alginolyticus* (WP_005373349.1) and the other from *V. coralliilyticus* (WP_006961879.1), share a highly similar N-terminus (65% identity) and a less similar C-terminus (37% identity). Furthermore, their downstream-encoded immunity proteins are not similar (Fig 1A). This observation led us to hypothesize that the highly similar N-termini of these effectors contain a previously undescribed T6SS secretion domain.

Following these observations, we set out to identify sequences homologous to the N-terminus of the *V. alginolyticus* effector in other bacterial proteins. Computational analyses revealed 4,434 unique protein accession numbers encoded by 12,071 genomic loci spread across various Gram-negative bacterial orders, such as Enterobacterales, Burkholderiales, Pseudomonadales, and Aeromonadales (Fig 1B and S1 Dataset). Notably, >98% of the genomes encoding these homologs contain a T6SS (S2 Dataset).

A multiple sequence alignment of the homologous sequences revealed a bipartite conserved motif. The first region corresponds to amino acids 31–126 and the second corresponds to amino acids 271–476 in the *V. alginolyticus* effector (Fig 1C). We named this domain WHIX

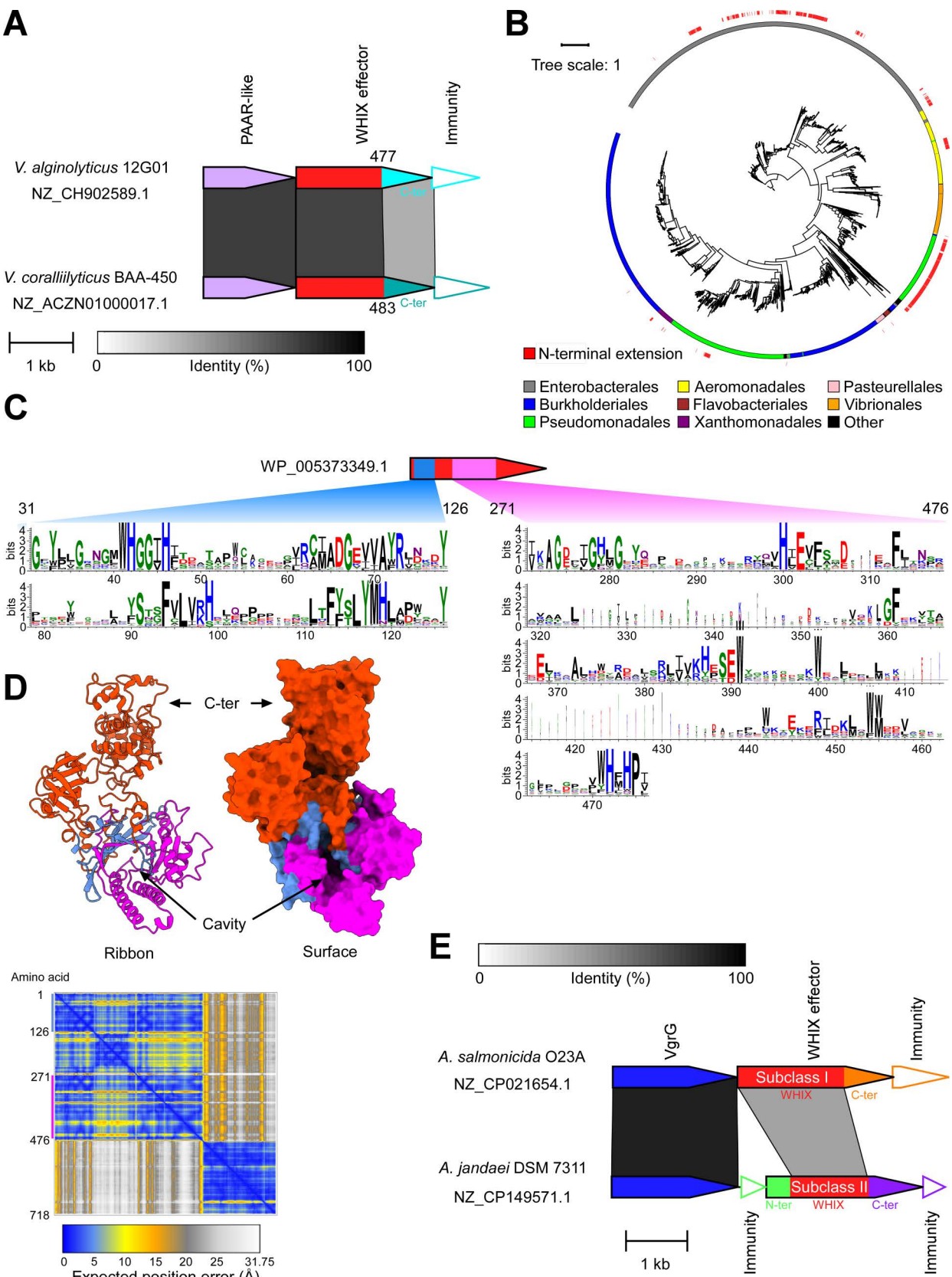

**Fig 1. WHIX domain defines a widespread family of T6SS effectors. (A)** Gene synteny of the T6SS effectors WP_005373349.1 from *Vibrio alginolyticus* and WP_006961879.1 from *Vibrio coralliilyticus*. Gray rectangles denote amino acid identity between regions. Bacterial strain and

Genbank accession are denoted. C-ter, C-terminus. **(B)** Phylogenetic distribution of WHIX domain-containing proteins. The bacterial order and the WHIX subclass are denoted by color. The evolutionary history was inferred using the Maximum Likelihood method and JTT + I + G4 model. The tree is drawn to scale, with branch lengths measured in the number of substitutions per site. Evolutionary analyses were conducted using IQ-TREE. The data underlying this figure can be found in S1 Data. **(C)** Conservation logo based on a multiple sequence alignment of WHIX domain sequences. The position numbers correspond to the amino acids in WP_005373349.1. **(D)** An AlphaFold 3 structure prediction of WP_005373349.1, as ribbon and surface representations. The first region of WHIX is colored blue; the second region is colored magenta. The AlphaFold Predicted Aligned Error (PAE) plot is shown below. The color key represents the expected position error for each pair of residues in Å units. Blue represents low predicted errors, indicating high confidence in the relative positions of those residues. The orange color indicates higher predicted errors, suggesting lower confidence. C-ter, C-terminus. The data underlying this figure can be found in S1 File. **(E)** Gene synteny of representative subclass I (top) and II (bottom) WHIX effectors, WP_087755869.1 from *A. salmonicida* O23A and WP_082035413.1 from *A. jandaei* DSM 7311, respectively. Gray rectangles denote amino acid identity between regions. Bacterial strain and Genbank accession are denoted. N-ter, N-terminal extension; C-ter, C-terminal extension.

after a conserved WHxxxH motif found in the first region (WH type sIX motif). The predicted structure of the *V. alginolyticus* effector, generated with AlphaFold 3 [53], suggests that the bipartite WHIX motif forms a distinct folded module with a deep cavity (Fig 1D and S1 File). The predicted C-terminal toxic domain appears to form a separate module.

Analysis of WHIX-containing proteins revealed two subclasses (Fig 1B and 1E). In subclass I (e.g., WP_087755869.1), the WHIX domain is located at the N-terminus whereas in subclass II (~16% of the unique protein accession numbers; e.g., WP_082035413.1), an additional extension, >99 amino acids long, is found N-terminal to WHIX. Subclass II members reside predominantly in Enterobacterales, Pseudomonadales, and Aeromonadales genomes (Fig 1B).

Most WHIX-containing proteins (96.5% of the unique protein accession numbers) contain a C-terminal extension. When examining the identity of these C-terminal extensions, we found diverse domains predicted to predominantly target the peptidoglycan. These include for example, Glycosyl hydrolase 19, Chitinase-like, Pesticin lysozyme-like, Lambda lysozyme-like, and Beta- N-acetylglucosaminidase (Table 1 and S1 Dataset and S1A Fig and S2 File). In instances in which WHIX-containing proteins lack a C-terminal extension, they can be found upstream of genes encoding a predicted peptidoglycan-degrading enzyme (e.g., WP_227130466.1 and WP_223384031.1). Analyses of the N-terminal extensions found in subclass II WHIX-containing proteins suggested that they also comprise diverse peptidoglycan-targeting enzymes, such as Tae3-like, M23 superfamily, NlpC_P60-like, Peptidase M15-like, and DL-endopeptidase-like (Table 2 and S1 Dataset and S1B Fig and S3 File); in two instances, WHIX is fused to an N-terminal VgrG domain. Further inspection of the genes immediately downstream and upstream of the predicted toxic domains fused to WHIX revealed that the vast majority encode proteins with a predicted signal peptide (S1 Dataset), indicating possible periplasmic localization. Moreover, some of these proteins contain domains previously implicated in toxin antagonism and T6SS immunity, such as the DUF1311/LrpI lysozyme inhibitor [54,55] and DUF3828/Tai3 [56] (S1 Dataset), suggesting that the genes flanking the WHIX-encoding gene encode the cognate immunity proteins to the adjacently-encoded toxic domains. Taken together, the above observations led us to conclude that WHIX defines a new class of polymorphic antibacterial T6SS effectors.

## The *Aeromonas jandaei* T6SS is an active antibacterial system

To further investigate WHIX effectors, we characterized the T6SS in *Aeromonas jandaei* DSM 7311 (also named CECT 4228 and ATCC 49568), which is predicted to encode a subclass II WHIX effector (S1 Dataset) that we reasoned can be used as a model effector. Since the publicly available genome of this strain was not assembled into a chromosome (NCBI RefSeq assembly GCF_000819955.1), we re-sequenced the genome, resulting in a complete chromosome assembly (NCBI Reference Sequence: NZ_CP149571.1). Analysis of the newly

**Table 1. Domains and activities identified at C-termini of WHIX effectors.**

| Number of unique proteins | Predicted domain | Example accession numbers |
|---|---|---|
| 2,242 | Chitinase-like[a,b] | WP_235364548.1, WP_005532457.1, WP_254694357.1, WP_061356769.1, WP_047482042.1, WP_242462213.1, WP_256859882.1, WP_157640791.1, WP_058834442.1, WP_158661206.1, WP_190283627.1, WP_248215477.1 |
| 448 | Pesticin lysozyme-like[b] | WP_152561995.1, WP_252037023.1, WP_229692790.1, WP_197970291.1 |
| 432 | Lysozyme-like[b] | WP_267196538.1, WP_032211049.1, WP_274181215.1 |
| 353 | N-acetylmuramidase[a,b] | WP_000125544.1, WP_244581361.1, WP_175870885.1, WP_200891482.1 |
| 346 | N-acetylglucosaminidase[a,b] | WP_167852933.1, WP_135408874.1, WP_226908484.1 |
| 111 | PG hydrolase FlgJ-like[b] | WP_240725722.1 |
| 101 | Chitosanases-like[a] | WP_023247212.1 |
| 43 | Cell wall hydrolase[b] | WP_186012365.1 |
| 22 | NlpC_P60-like[b] | WP_142392822.1 |
| 12 | Lytic transglycosylase-like[b] | WP_019466345.1 |
| 11 | Colicin pore-forming[a] | WP_115426702.1, WP_136039692.1 |
| 4 | Adenylyl cyclase toxin-like[a] | WP_097513094.1 |
| 145 | Others | |
| 163 | No domain/truncated | |

[a]Identified using HHpred analyses.

[b]Identified using the NCBI CDD.

**Table 2. Domains and activities identified at N-termini of subclass II WHIX effectors.**

| Number of unique proteins | Predicted domain | Example accession number |
|---|---|---|
| 328 | Tae3-like[a] | WP_282812471.1 |
| 222 | NlpC_P60-like[b] | WP_205350834.1 |
| 162 | M23 superfamily[b] | WP_069143508.1 |
| 13 | MepA-like DD-peptidase[a] | WP_158695274.1 |
| 5 | Lysozyme-like[b] | WP_281319639.1 |
| 4 | DL-endopeptidase-like[a] | WP_267540890.1 |
| 2 | Peptidase M15-like[b] | WP_206728780.1 |
| 2 | VgrG[b] | WP_062088695.1 |
| 1 | AmpD-like amidase[b] | WP_044238255.1 |
| 1 | PGRP[b] | WP_181197892.1 |
| 1 | PG_binding1 + Peptidase M15[a,b] | WP_266216180.1 |
| 1 | Various[b] | WP_284721687.1 |

[a]Identified using HHpred analyses.

[b]Identified using the NCBI CDD.

assembled genome revealed a main T6SS gene cluster with a predicted Rhs-containing effector similar to the previously reported TseI [57], and two auxiliary modules containing T6SS tube-spike components: one encoding a Tle1-like effector [58] and the other a subclass II WHIX effector, WP_082035413.1, which we named Awe1 (*Aeromonas* WHIX effector 1) (S2A Fig). Notably, the identification of another T6SS operon containing a DUF3289-encoding gene will be described below.

To identify the conditions in which this T6SS is active, we monitored the expression and secretion of the hallmark secreted T6SS component, Hcp [14], using a custom-made antibody

designed to detect all three Hcp proteins encoded by *A. jandaei* (S2A Fig). As shown in S2B Fig, the T6SS is active between 23 and 37 °C, in media containing either 1% or 3% (wt/vol) NaCl (LB or MLB media, respectively). Hcp secretion is T6SS-dependent, since it was not observed in a T6SS⁻ strain in which we deleted the core component *tssB*. Secretion of Hcp was restored when *tssB* was complemented from a plasmid (S2C Fig).

The effectors that we computationally identified in *A. jandaei* are encoded next to putative immunity genes (S2A Fig). Therefore, we hypothesized that this T6SS plays a role in inter-bacterial competition. To test this hypothesis, we used the wild-type *A. jandaei* and its T6SS⁻ mutant strain (Δ*tssB*) as attackers in competition against *Escherichia coli* MG1655 prey. Both the wild-type and the *tssB*-complemented mutant attacker strains, but not the T6SS⁻ mutant containing an empty plasmid, killed the *E. coli* prey during a 4-h co-incubation, as evidenced by a decrease in prey viability; although toxicity was apparent under all tested temperature and salinity conditions, it was significantly more pronounced in low salt media (LB; 1% [wt/vol] NaCl) compared to high salt media (MLB; 3% [wt/vol] NaCl) (S2D Fig). Based on these findings, we performed subsequent assays in media containing 1% [wt/vol] NaCl (LB) at 30 °C. Taken together, our results indicate that the T6SS in *A. jandaei* is active under a wide range of growth conditions in which it mediates antibacterial toxicity.

### *Aeromonas jandaei* T6SS secretes at least four antibacterial effectors

To characterize the T6SS effector repertoire of *A. jandaei*, we next employed a comparative proteomics approach. Using mass spectrometry, we compared the proteins secreted by the wild-type *A. jandaei* (T6SS⁺) with those secreted by a Δ*tssB* mutant strain (T6SS⁻) when grown in LB media at 30 °C. We identified 13 proteins significantly enriched in the supernatant of the wild-type strain (S2E Fig and S4 File). These include the four structural VgrG proteins (note that VgrG2 is separated into 'a' and 'b' because the mass spectrometry data were analyzed against a genome annotation in which the gene is split into two open reading frames; S4 File), the three structural Hcp proteins (note that Hcp1 and Hcp3 are identical and are thus annotated together), and the three computationally predicted effectors (TseI, Tle1, and Awe1). In addition, we identified a putative effector containing DUF3289 (WP_042032936.1), which was recently proposed to be an antibacterial toxic domain that targets the bacterial periplasm [48,59,60] (S2A Fig). Although not annotated in the NCBI RefSeq genome, we identified a short open reading frame immediately downstream of the DUF3289-encoding gene (between position 2,059,343 and 2,059,057 in genome NZ_CP149571.1), encoding a 95 amino acids long hypothetical protein with an N-terminal signal peptide and three transmembrane helices (analyzed using Phobius [61]) (S2A Fig). We hypothesized that this is an immunity protein antagonizing the DUF3289-containing effector. We also identified three flagella proteins (FlaB, FlgM, and FliD) as preferentially found in the wild-type strain's secretome; notably, manipulation of the T6SS was previously reported to affect motility in other *Aeromonas* species [62,63].

If TseI, Tle1, and DUF3289 are T6SS effectors and their downstream genes encode cognate immunity proteins, then deletion of each predicted effector and immunity pair should render a prey strain sensitive to attack by a wild-type *A. jandaei* attacker that delivers the effector via its T6SS, and not by a T6SS⁻ mutant strain (Δ*tssB*). Indeed, we found that deleting these gene pairs renders prey strains sensitive to a T6SS-mediated attack during self-competition assays, as evidenced by the decrease in prey viability over time (S3 Fig). Importantly, the expression of the predicted immunity genes from a plasmid restored the prey strains' ability to antagonize the T6SS-mediated attack. Taken together, these results confirm that TseI, Tle1, and DUF3289 are antibacterial T6SS effectors. The fourth effector, Awe1, is the focus of subsequent sections.

## WHIX is sufficient for T6SS-mediated secretion of multiple domains

Next, we focused our investigation on Awe1, a subclass II WHIX effector predicted to contain two toxic domains, one on each side of WHIX (Figs 1E and 2A). Prediction of the Awe1 structure using AlphaFold 3 revealed a similar fold to that of WP_005373349.1 from *V. alginolyticus*, containing a cavity made by a bipartite WHIX domain (Fig 2B and S5 File). In addition, two domains, one at the N-terminus and one at the C-terminus of Awe1, were apparent in the prediction. Furthermore, when the Awe1 structure was predicted in a heterotrimer together with the putative immunity proteins encoded upstream (AwiU) and downstream (AwiD) of Awe1 (Fig 2A), we found that these immunity proteins are expected to bind the N-terminal and C-terminal domains of Awe1, respectively (Fig 2C and S6 File). Indeed, this prediction was confirmed in a bacterial two-hybrid assay in which we found that AwiU binds the N-terminal domain of Awe1 whereas AwiD binds the C-terminal domain of Awe1 (Fig 2D). These results imply that the N- and C-termini extensions fused to WHIX are toxic domains.

The N-terminal domain of Awe1 (amino acids 1–147) is predicted to belong to the EnvC superfamily (according to NCBI conserved domain database; CDD [64]), which includes peptidoglycan peptidase domains [65]. In agreement, the AlphaFold 3-predicted structure of this domain is similar to the M23 peptidase domain of the ShyA endopeptidase from *Vibrio cholerae* (PDB: 6UE4A), as determined by Foldseek analysis [66] (S1 Dataset and S4A Fig and S7 File). In contrast, no similarity to known domains was identified for the Awe1 C-terminal domain (amino acids 700–862) by sequence homology, although analysis of members of the C-terminal domain cluster to which it belongs suggests that it is a Beta-N-acetylglucosaminidase (S1 Dataset). Nevertheless, the AlphaFold 3-predicted structure of this domain was found to share low similarity with the predicted structure of the FlgJ peptidoglycan hydrolase from *Vibrio parahaemolyticus* (AlphaFold identifier: Q9X9J3), as determined by Foldseek analysis (S4B Fig and S7 File). Based on these observations, we hypothesized that WHIX is a T6SS secretion domain that can carry two toxic domains, one on each of its ends.

To determine whether the Awe1 WHIX domain is sufficient to mediate T6SS-dependent secretion, we monitored the secretion of plasmid-expressed full-length Awe1 fused to a C-terminal Myc tag, or its N- and C-terminal truncated forms (Fig 2E) from *A. jandaei* in which we deleted the genomic *awe1* (Δ*awe1*; T6SS$^+$) or from a derivative in which we inactivated the T6SS (Δ*awe1*/ Δ*tssB*; T6SS$^-$). Truncation of the Awe1 N-terminal domain (Awe1$^{148-862}$), C-terminal domain (Awe1$^{1-699}$), or both domains together (Awe1$^{148-699}$), did not hamper the T6SS-dependent secretion of Awe1 (Fig 2F), indicating that the WHIX domain alone is sufficient for T6SS secretion.

To further support WHIX's role as a T6SS secretion domain that can carry two domains, one on each end, we monitored the secretion of Awe1 in which the N-terminal domain was swapped with a glutathione S-transferase (GST) domain and the C-terminal domain was swapped with a superfolder green fluorescent protein (GFP) (Fig 2E). As shown in Fig 2G, the Awe1 WHIX domain (amino acids 148–699) could carry two non-T6SS domains, fused to its N- and C-terminal ends individually or together, for secretion via the T6SS. These results implicate WHIX as a T6SS secretion domain able to mediate the secretion of two flanking domains.

## Awe1 is a double-blade effector

The findings described above support the hypothesis that Awe1 contains two distinct toxic domains and is thus akin to a double-blade sword. Based on this hypothesis, we first sought to determine whether the two distinct toxic domains in Awe1 are functional. Using *E. coli* as a surrogate bacterial cell, we found that Awe1 is toxic when sent to the bacterial periplasm (by

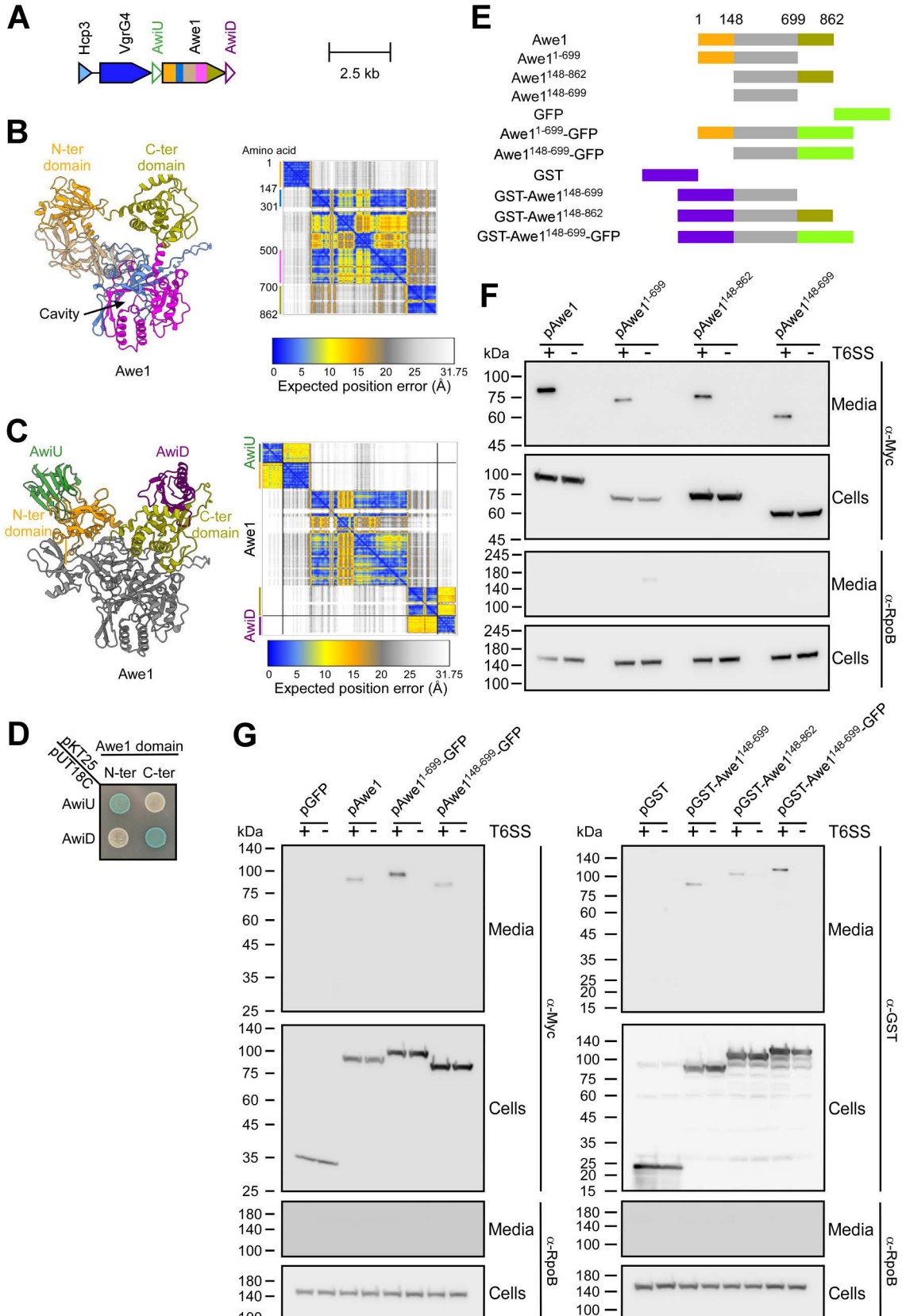

**Fig 2. WHIX can carry two flanking domains for secretion via the T6SS. (A)** Schematic representation of the Awe1-encoding auxiliary T6SS operon in *Aeromonas jandaei* DSM 7311. The colors in Awe1 correspond to the domain colors in panel B. **(B and C)**

AlphaFold 3 structure predictions of Awe1 (B) or Awe1 in complex with AwiU and AwiD lacking their predicted N-terminal signal peptides (C). The first part of the Awe1 WHIX domain is colored blue (amino acids 148–301); the second part of WHIX (amino acids 500–699) is colored magenta; the N-terminal (N-ter) Awe1 domain fused to WHIX (amino acids 1–147) is colored orange; the C-terminal (C-ter) Awe1 domain fused to WHIX (amino acids 700–862) is colored beige; AwiU is colored green; AwiD is colored purple. The AlphaFold Predicted Aligned Error (PAE) plot is also shown; black lines denote the borders between proteins included in the prediction. The color key represents the expected position error for each pair of residues in Å units. Blue represents low predicted errors, indicating high confidence in the relative positions of those residues. The orange color indicates higher predicted errors, suggesting lower confidence. The data underlying this figure can be found in S5 File (A) and S6 File (B). **(D)** Bacterial two-hybrid assay in which the indicated proteins were expressed in an *Escherichia coli* BTH101 reporter strain fused to the T18 or T25 domain of the *Bordetella* adenylate cyclase. Bacteria were spotted on plates supplemented with the chromogenic substrate X-Gal. A blue-colored colony indicates protein-protein interaction. Awe1 N-ter and C-ter are the same as detailed above; AwiU and AwiD were subcloned without their predicted N-terminal signal peptides. A representative result out of three independent experiments is shown. **(E)** Schematic representation of the Awe1 truncations and chimeras used in the secretion assays shown in panels F and G. **(F** and **G)** Expression (cells) and secretion (media) of the indicated C-terminally Myc-tagged or N-terminally GST-tagged Awe1 forms expressed from an arabinose-inducible plasmid (pAwe1) in Δ*awe1 A. jandaei DSM 7311* (T6SS⁺) or Δ*awe1*/Δ*tssB* (T6SS⁻) mutant strains grown for 3 h at 30 °C in LB media supplemented with kanamycin and 0.05% ʟ-arabinose. GFP, superfolder GFP. RNA polymerase beta subunit (RpoB) was used as a loading and lysis control. Results from a representative experiment out of at least three independent experiments are shown.

fusing it to an N-terminal PelB signal peptide) (Figs 3A and S5). In support of the hypothesis that both the N- and C-terminal domains of Awe1 are functional toxic domains, we observed that expression of either is detrimental to *E. coli* when sent to the periplasm. Notably, the periplasmic expression of the WHIX domain alone is not toxic to *E. coli*, and its effect on *E. coli* growth is comparable to that of the full-length Awe1 expressed in the cytoplasm.

Next, we sought to determine whether the two Awe1 toxic domains require two distinct cognate immunity proteins (i.e., AwiU and AwiD) to antagonize them. To this end, we generated an *A. jandaei* strain in which we deleted the genes encoding the predicted effector Awe1 and both predicted immunity proteins AwiU and AwiD (ΔI-E-I), and we used it as prey in self-competition assays. This strain was killed by a wild-type *A. jandaei* attacker during competition, as evidenced by the decrease in viability over 4 h of co-incubation on an LB agar plate (Fig 3B). The killing was mediated by the T6SS-delivered Awe1, since inactivation of the T6SS (Δ*tssB*) or Awe1(Δ*awe1*) enabled prey growth during the co-incubation period. Expression of the full-length Awe1 from a plasmid in the Δ*awe1* attacker strain restored its ability to kill the sensitive prey, as did the expression of an Awe1 truncated version containing only the WHIX domain fused to the C-terminal toxic domain (Awe1$^{148–862}$). Expression of AwiD from a plasmid in the prey restored its ability to antagonize both full-length Awe1 and its truncated version lacking the predicted N-terminal toxic domain (Awe1$^{148–862}$). Surprisingly, however, the expression of an Awe1 truncation containing only the WHIX domain fused to the N-terminal toxic domain (Awe1$^{1–699}$) was not toxic towards the prey strain, and the AwiU predicted immunity protein did not antagonize Awe1-mediated toxicity during co-incubation (Fig 3B). These results suggest that although the C-terminal toxic domain of Awe1 is functional during self-competition and is antagonized by the downstream encoded AwiD, the predicted N-terminal toxic domain of Awe1 is not toxic under these conditions.

Because the N-terminal domain of Awe1 is toxic when expressed in *E. coli* (Fig 3A), we hypothesized that although it is a functional toxic domain, a yet-unknown non-immunity protein-resistance mechanism abrogates its toxicity toward *Aeromonas* in self-competition assays. A similar observation was made with the peptidoglycan-targeting effector TseH from *V. cholerae* [67]. Therefore, we set out to determine whether the N-terminal Awe1 domain can intoxicate an *E. coli* prey in a T6SS-dependent manner and whether AwiU can antagonize it. To this end, we first constructed an *A. jandaei* mutant strain in which we deleted the four known effectors (*Aj*$^{effectorless}$) to obtain an effectorless strain unable to intoxicate *E. coli*

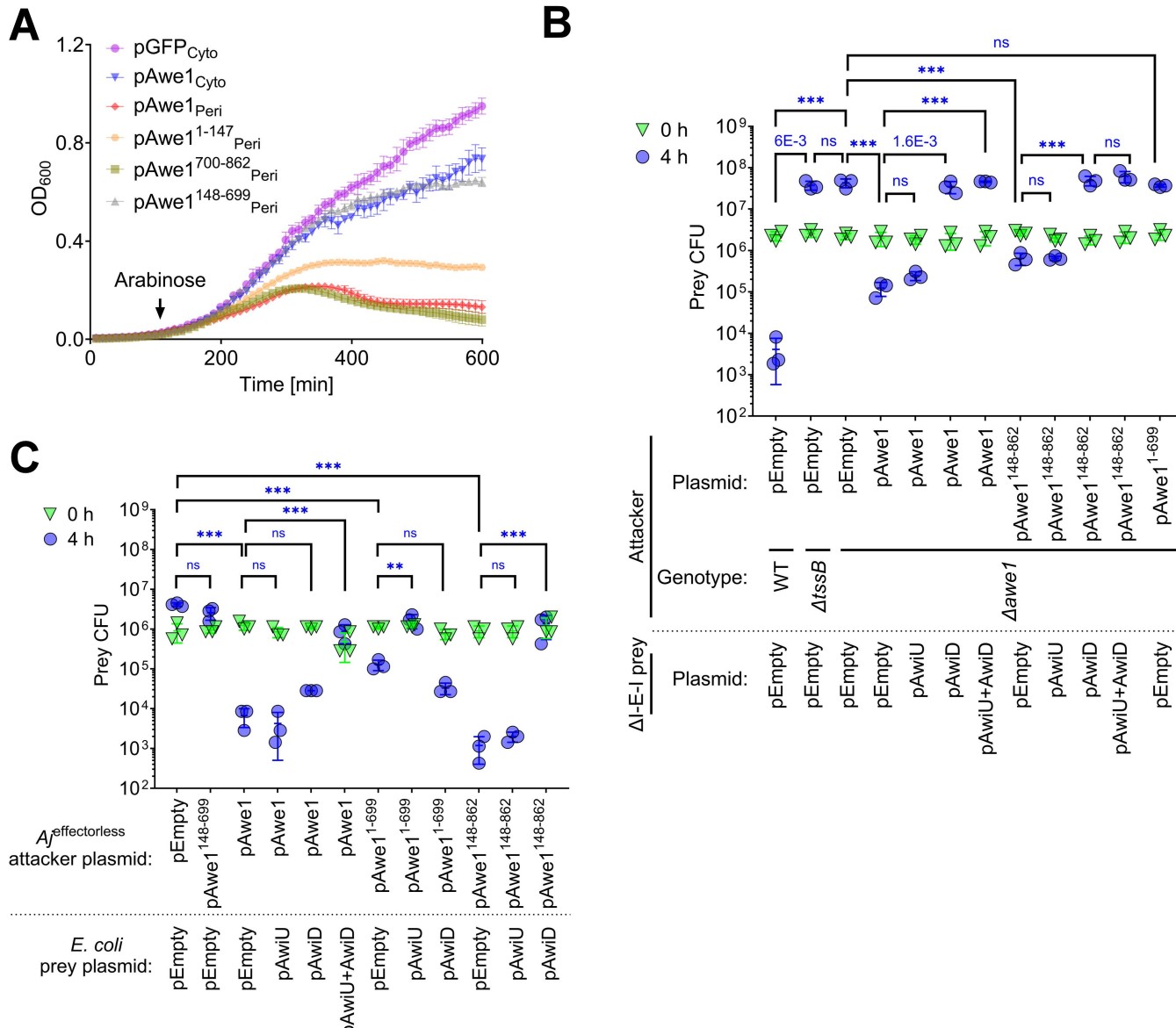

**Fig 3. Awe1 is a double-blade effector. (A)** Growth of *Escherichia coli* MG1655 containing plasmids for the arabinose-inducible expression of superfolder GFP (GFP) or the indicated Awe1 forms. An arrow denotes the time point at which L-arabinose (0.05% [wt/vol]) was added to the LB media. Cyto, cytoplasmic expression; Peri, periplasmic expression by fusing the protein to an N-terminal PelB signal peptide. Data are shown as the mean ± SD; *n* = 4 biological samples. The experiments were repeated three times with similar results. Results from representative experiments are shown. **(B** and **C)** Viability counts (CFU) of *Aeromonas jandaei* DSM 7311 prey strains in which we deleted the genes encoding AwiU, Awe1, and AwiD (ΔI-E-I) (B) or of *E. coli* BL21(DE3) prey strains (C) containing an empty plasmid (pEmpty) or a plasmid for the arabinose-inducible expression of AwiU (pAwiU), AwiD (pAwiD), or both (pAwiU+AwiD) before (0 h) and after (4 h) co-incubation with the indicated *A. jandaei* DSM 7311 attacker strains containing an empty plasmid or a plasmid for the arabinose-inducible expression of the indicated Awe1 form, on LB plates supplemented with 0.05% (in B) or 0.1% (in C) (wt/vol) L-arabinose at 30 °C. The statistical significance between samples at the 4-h time point was calculated using one-way ANOVA with Tukey multiple comparisons test on log-transformed data; ***$P$< 0.0001; **$P$ < 0.0003; ns, no significant difference ($P$ > 0.05); WT, wild-type; DL, the assay's detection limit. Data are shown as the mean ± SD; *n* = 3. The data shown are a representative experiment out of at least three independent experiments. The data underlying this figure can be found in S2 Data.

(Fig 3C). Expression of the full-length Awe1 from a plasmid in the effectorless attacker strain restored its ability to kill *E. coli* prey. In contrast to the results obtained in the *Aeromonas* self-competition assay (Fig 3B), AwiD alone could not antagonize the full-length Awe1-mediated

toxicity when expressed from a plasmid in the *E. coli* prey strain (Fig 3C). AwiU alone was also unable to antagonize the full-length Awe1-mediated toxicity. However, co-expression of AwiU and AwiD from a plasmid significantly restored the prey strain's ability to antagonize the attack. Furthermore, in contrast to the results obtained in the *Aeromonas* self-competition assay (Fig 3B), the expression of a truncated version of Awe1 containing only the N-terminal toxic domain and the WHIX domain (Awe1$^{1-699}$) intoxicated the *E. coli* prey strain, and the expression of AwiU alone in the prey strain, but not of AwiD, was sufficient to antagonize this toxicity (Fig 3C). Taken together, these results confirm our hypothesis that Awe1 is a double-blade effector containing two distinct toxic domains, one on each end of the WHIX domain, and requiring two cognate immunity proteins to antagonize their toxicity.

### The C-terminus of VgrG4 is required for Awe1 delivery

Many T6SS cargo effectors were shown to be loaded onto a secreted T6SS tube-spike core component encoded next to them on the genome. Therefore, we hypothesized that Awe1 is loaded onto the upstream-encoded secreted spike component VgrG4 (WP_198493475.1) for delivery via the T6SS. In agreement with our findings that the Awe1 WHIX domain is sufficient for T6SS-dependent secretion, an AlphaFold structure prediction implies that Awe1 is loaded onto VgrG4; the predicted structure suggests that the deep cavity of the WHIX domain caps the C-terminal tail of VgrG4 (Fig 4A and S8 File). In further support of our hypothesis, the full-length VgrG4 co-precipitates with the WHIX domain of Awe1 (Awe1$^{148-699}$) while a VgrG4 truncated version lacking the 14 C-terminal amino acids (VgrG4$^{1-680}$) does not (Fig 4B). These results indicate that Awe1 binds VgrG4 and that the WHIX domain and the C-terminal end of VgrG4 mediate the interaction.

To determine whether the C-terminus of VgrG4 is required for Awe1 delivery into a prey cell, we constructed a strain in which we deleted *vgrG4*. When this Δ*vgrG4* strain was used as an attacker in a self-competition assay against a ΔI-E-I prey strain (specifically sensitive to intoxication by Awe1), it was unable to intoxicate the prey, similar to the T6SS⁻ attacker (Δ*tssB*) (Fig 4C). Importantly, *vgrG4* deletion did not hamper overall T6SS activity, since this mutant can secrete Hcp and intoxicate an *E. coli* prey strain (S6 Fig). These results indicate that VgrG4 is specifically required for Awe1 delivery. Moreover, although expression of the full-length VgrG4 from an arabinose-inducible plasmid restored antibacterial activity to the Δ*vgrG4* attacker strain in self-competition assays, expression of VgrG4 in which we truncated the 14 C-terminal amino acids (VgrG4$^{1-680}$) was unable to restore toxicity (Fig 4C). Notably, although both the full-length and truncated VgrG4 are secreted in a T6SS-dependent manner, the Awe1 WHIX domain (Awe1$^{148-699}$) is only secreted to the medium in the presence of the full-length VgrG4 (Fig 4D). This implies that the absence of Awe1-mediated intoxication in the presence of the truncated VgrG4 resulted from the effector's inability to properly load onto the spike and be secreted, not from hampered VgrG4 secretion. Taken together, these results indicate that the Awe1 WHIX domain is loaded onto the C-terminus of VgrG4 for T6SS-mediated secretion.

### Discussion

In this work, we describe a widespread class of polymorphic T6SS effectors that share a domain named WHIX. We show that the WHIX domain is sufficient to mediate secretion via the T6SS, possibly by loading onto a C-terminal tail of a secreted spike component. Importantly, we demonstrate a unique trait of WHIX domains—they are found in classic effectors containing one toxic domain, as well as in effectors akin to a double-blade sword containing two distinct toxic domains, each requiring its own cognate immunity protein.

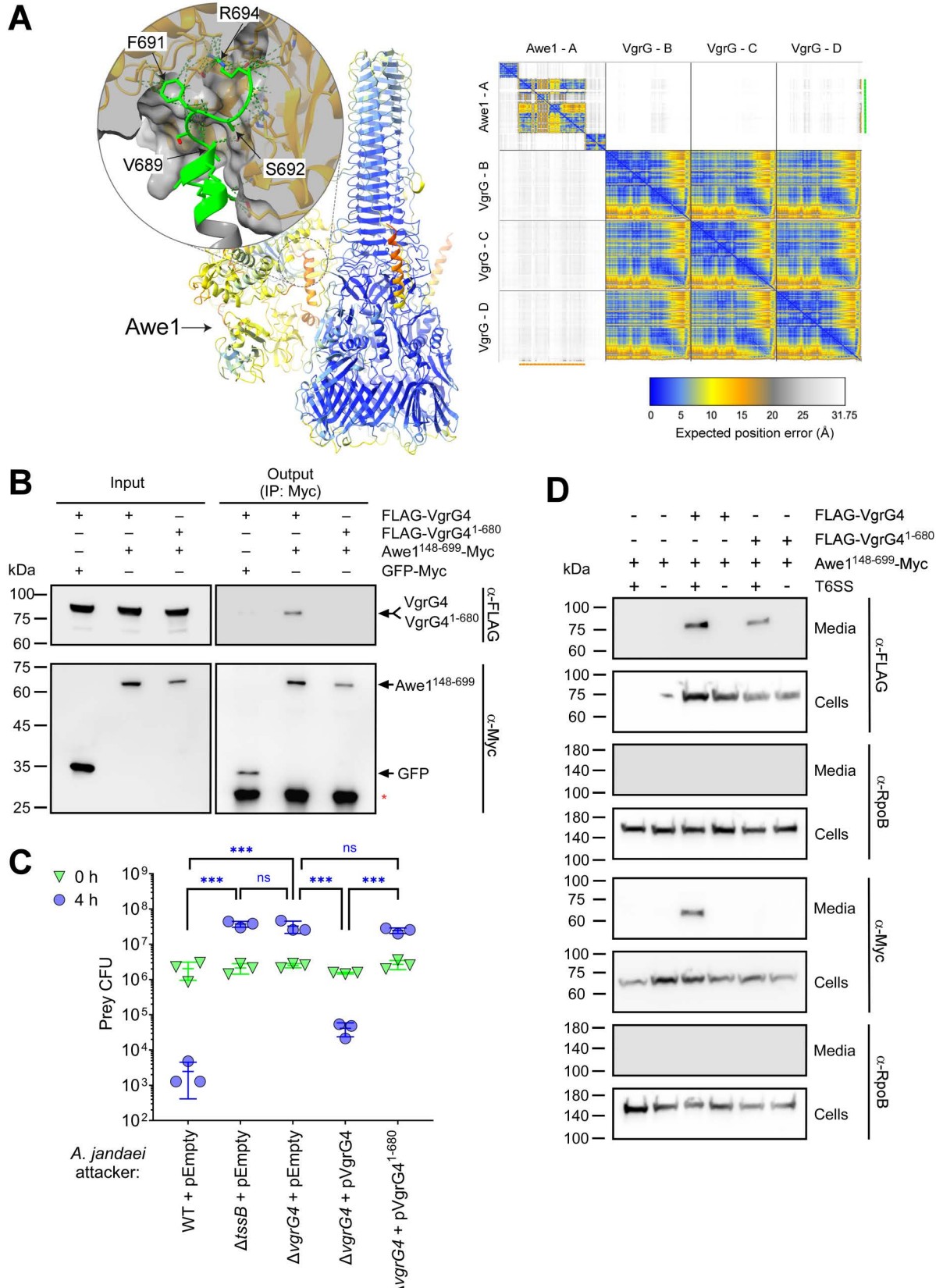

**Fig 4. The C-terminus of VgrG4 is required for Awe1 delivery. (A)** AlphaFold structure prediction of the complex assembled by an Awe1 monomer and a VgrG4 trimer, shown as a ribbon representation. The color gradient from blue to orange correlates with the

Predicted Aligned Error (PAE) values, where blue represents regions with a very high prediction confidence level, and orange represents regions with a very low confidence level. The inset is a close-up view of the predicted Awe1-VgrG4 interacting region. VgrG4 and Awe1 residues predicted to interact with each other are represented in green and orange, respectively. Predicted interactions are shown with dotted lines. The C-terminal residues of VgrG4: Lys684, Ser685, Lys688, Val689, Ser690, Phe691, Ser692, Gly693, and Arg694 are predicted to interact with residues within the WHIX domain cavity of Awe1; for clarity, only four VgrG4 residues are labeled. The AlphaFold Predicted Aligned Error plot is shown on the right. The color key represents the expected position error for each pair of residues in Å units. Blue represents low predicted errors, indicating high confidence in the relative positions of those residues. The orange color indicates higher predicted errors, suggesting lower confidence. Awe1 (chain A) and VgrG4 (chain D) residues interacting are underscored with orange (x-axis) and green lines (y-axis), respectively. These regions correspond to the 11 C-terminal residues of VgrG4 and the WHIX domain of Awe1, respectively. **(B)** Immunoprecipitation using α-Myc antibodies from *Escherichia coli* BL21 (DE3) strains co-expressing the indicated C-terminal Myc-tagged and N-terminal FLAG-tagged proteins from arabinose-inducible plasmids. A red asterisk denotes the short chain of the antibody used for precipitation. **(C)** Viability counts (CFU) of *Aeromonas jandaei* DSM 7311 prey strains in which we deleted the genes encoding AwiU, Awe1, and AwiD (ΔI-E-I) before (0 h) and after (4 h) co-incubation with the indicated *A. jandaei* DSM 7311 attacker strains containing an empty plasmid (pEmpty) or a plasmid for the arabinose-inducible expression of the indicated N-terminally FLAG-tagged VgrG4 forms, on LB plates supplemented with 0.05% (wt/vol) L-arabinose at 30 °C. The statistical significance between samples at the 4-h time point was calculated using one-way ANOVA with Tukey multiple comparisons test on log-transformed data; ***$P < 0.0001$; ns, no significant difference ($P > 0.05$); WT, wild-type. Data are shown as the mean ± SD; $n = 3$. **(D)** Expression (cells) and secretion (media) of the N-terminally FLAG-tagged VgrG4 forms used in panel B and a C-terminally Myc-tagged Awe1 WHIX domain (Awe1$^{148–699}$), expressed from an arabinose-inducible in *A. jandaei* DSM 7311 Δ*vgrG4* (T6SS$^+$) or a T6SS$^-$ mutant strain (Δ*vgrG4*/Δ*tssB*) grown for 3 h at 30 °C in LB media supplemented with chloramphenicol and 0.005% (wt/vol) L-arabinose. RNA polymerase beta subunit (RpoB) was used as a loading and lysis control. In B–D, results from a representative experiment out of at least three independent experiments are shown. The data underlying this figure can be found in S3 Data.

WHIX domains are found almost exclusively in proteins encoded by bacterial genomes harboring a T6SS. Taken together with the proximity of many WHIX-encoding genes to T6SS tube-spike components (e.g., VgrG and PAAR; S1 Dataset) and the fact that WHIX-containing proteins have been demonstrated to secrete via the T6SS in bacteria from different orders (i.e., Vibrionales and Aeromonadales) [21,52,68], we conclude that WHIX-containing proteins are T6SS effectors.

We show that WHIX effectors can be divided into subclass I, in which WHIX is at the N-terminus and usually fused to a C-terminal toxic domain, and subclass II, containing two toxic domains fused to either side of WHIX. Bifunctional T6SS effectors containing two catalytic domains targeting the peptidoglycan were previously described by Le and colleagues [69]. However, in contrast to subclass II WHIX-effectors, these bifunctional effectors appear to require a single cognate immunity protein to antagonize their toxicity. Here, we demonstrate that subclass II WHIX effectors, akin to double-blade swords, contain two toxic domains that function separately, each requiring its own cognate immunity protein to prevent intoxication. Notably, Jiang and colleagues recently described an effector with two catalytic domains in *A. veronii* [70], which is a homolog of Awe1; however, misannotation of the upstream-encoded immunity protein as a DUF4123 family adaptor, in combination with investigating its toxicity only in self-competition assays, possibly prevented its identification as a double-blade effector.

We hypothesize that WHIX domains' ability to carry two fused toxic domains reduces the risk of resistance developing in prey strains [71], as well as expands the potential target range of a single effector. Indeed, we find that the N-terminal toxic domain of Awe1 does not intoxicate *Aeromonas* in self-competition, yet it is functional against *E. coli* prey in which both the downstream-encoded immunity protein and the upstream-encoded immunity protein are required to antagonize Awe1-mediated toxicity. It is possible that the N-terminal toxic domain targets a peptidoglycan component that differs between *Aeromonas* and other bacteria. Alternatively, two toxic domains may allow the effector to function under a wider range of conditions. For example, if the N-terminal and C-terminal toxic domains are active in the prey cell under different salinity or temperature conditions. Such conditional effector toxicity was previously reported in effectors of *Pseudomonas aeruginosa* [27]. Moreover, the unique

ability of the WHIX domain to carry two flanking domains, either toxic or non-T6SS related, during T6SS-mediated secretion makes it a promising chassis for the future engineering of wide-range antibacterial toxins.

WHIX is predicted to comprise a bipartite sequence that folds into a distinct domain containing a deep cavity. Based on our experimental findings and structure predictions, we propose that this cavity caps the C-terminal tail of a secreted spike component (e.g., VgrG), constituting a unique loading mechanism of T6SS effectors onto the tube-spike complex. In future work, we will determine the structure of WHIX domains and their mechanism of loading onto the secreted T6SS tube-spike. In addition, the role and identity of sequences separating the two regions of the WHIX domain remain to be investigated.

Interestingly, WHIX domains appear to specifically reside within effectors predicted to target the peptidoglycan (with a handful of exceptions), as indicated by the annotation of domains fused to WHIX. Such specificity of toxic domains to a cellular target was previously reported for T6SS effectors containing a FIX domain, although FIX domains are exclusively fused to toxic domains that target the bacterial cytoplasm (e.g., nucleases) [32]. Such target specificity implies that WHIX may play a role in the peptidoglycan-targeting activity of the effector, although WHIX itself was not toxic when ectopically expressed in the *E. coli* periplasm. Even though we showed that WHIX is sufficient for T6SS-mediated secretion, we cannot rule out the possibility that it also plays an active role, perhaps after deployment of the effector within the recipient periplasm. In future work, we plan to investigate the mechanisms of action and the target of newly identified toxic domains fused to WHIX.

We also characterize the *A. jandaei* T6SS in this work. We demonstrate that this T6SS is active under a wide range of temperatures and medium salinity in which it mediates antibacterial activity using at least four effectors. Although three of these effectors are encoded adjacent to secreted core components of the T6SS, either within the main T6SS gene cluster or within T6SS auxiliary operons, the fourth effector is orphan. We demonstrate that this effector, containing a DUF3289 domain recently predicted to be similar to colicin M [60,72], is part of a *bona fide* T6SS effector and immunity pair. Previous works investigating the T6SS effector repertoires of *Aeromonas* strains focused only on effectors encoded within auxiliary modules containing secreted tube-spike components [58,70]. Therefore, it will be interesting to determine whether other *Aeromonas* strains, besides *A. jandaei*, also carry orphan T6SS effectors. In addition, analyses of T6SS effector repertoires in other bacteria, such as *V. parahaemolyticus* [73] and *V. coralliilyticus* [21] revealed core and accessory T6SS effector repertoires. Once additional information on orphan T6SS effectors is available in *Aeromonas*, a similar analysis can be performed on their T6SS effector repertoires as well.

In conclusion, we describe a class of polymorphic T6SS cargo effectors defined by a unique secretion domain—WHIX. The presence of WHIX in double-blade effectors reveals a new mechanism bacteria use to diversify their T6SS toxic effector payload. Our finding that WHIX effectors predominantly target the peptidoglycan raises questions regarding the possibility of T6SS secretion domains also playing an active role in the activity or localization of the effectors after deployment inside the recipient cell.

## Materials and methods

### Strains and media

For a complete list of strains used in this study, see S1 Table. *A. jandaei* DSM 7311 and its derivative strains were grown in lysogeny broth (LB; 1% [wt/vol] tryptone, 0.5% [wt/vol] yeast extract, and 1% [wt/vol] NaCl) or LB agar (1.5% [w/v]) plates at 30 °C. *E. coli* strains were grown in 2xYT broth (1.6% [wt/vol] tryptone, 1% [wt/vol] yeast extract, and 0.5% [wt/

vol] NaCl) at 37 °C, or at 30 °C when harboring effector-expressing plasmids. In cases where *A. jandaei* or *E. coli* contains a plasmid, the media were supplemented with chloramphenicol (10 μg/mL), ampicillin (100 μg/mL), or kanamycin (30 μg/mL) to maintain the plasmid. To repress effector expression from the P*bad* promoter in *E. coli*, 0.4% D-glucose (wt/vol) was added to the media. To induce expression from the P*bad* promoter, L-arabinose (0.005%–0.1% [wt/vol]) was added to the media, as indicated.

## Plasmids construction

For a complete list of plasmids used in this study, see S2 Table. For expression in *A. jandaei* or *E. coli*, the coding sequence (CDS) of TssB (WP_033114640.1), Awe1 (WP_082035413.1) or its indicated truncations, AwiU (WP_042029887.1), AwiD (WP_156128651.1), DUF3289 (WP_042032936.1), the immunity protein encoded downstream of DUF3289 (encoded between position 2,059,343 and 2,059,057 in genome NZ_CP149571.1), the immunity protein of TseI (WP_042032020.1), the immunity protein of Tle1 (WP_042031148.1), and VgrG4 (WP_198493475.1) were PCR amplified from *A. jandaei* DSM 7311 genomic DNA; The CDS of superfolder GFP and GST were amplified from available plasmids. Amplicons were inserted into the multiple cloning site (MCS) of pBAD^K/*Myc*-His, pBAD33.1 (Addgene), pKT25, pUT18C, or their derivatives using the Gibson assembly method [74]. The resulting plasmids were introduced into *E. coli* DH5α (λ-pir) by electroporation, and later into *A. jandaei* via conjugation. Trans-conjugants were selected on LB agar plates supplemented with the appropriate antibiotics. Since *A. jandaei* DSM 7311 is naturally resistant to ampicillin, this antibiotic was added to the selection plates.

## Construction of deletion strains

To construct deletion strains, 1 kb sequences upstream and downstream of each gene or region to be deleted were cloned into the MCS of pDM4, a Cm^ROriR6K suicide plasmid [75]. The pDM4 constructs were transformed into *E. coli* DH5α (λ-pir) by electroporation, and then transferred into *A. jandaei* strains via conjugation. Trans-conjugants were selected on agar plates supplemented with chloramphenicol (10 μg/mL) and ampicillin (100 μg/mL). The resulting *trans*-conjugants were grown on LB agar plates containing 10% (wt/vol) sucrose to select for the loss of the *sacB*-containing plasmid. Deletions were confirmed by PCR.

## *Aeromonas* secretion assays

For secretion of Hcp, Awe1 and its derivatives, and VgrG4, *A. jandaei* strains were grown overnight in LB; appropriate antibiotics were included when required to maintain an expression plasmid. Bacterial cultures were normalized to $OD_{600} = 0.5$ in 5 mL LB supplemented with appropriate antibiotics and 0.005%–0.1% (wt/vol) L-arabinose as indicated when expression from an arabinose-inducible plasmid was required. Bacterial cultures were incubated with constant shaking (220 rpm) for 3 h at 30 °C. After 3 h, 0.5 $OD_{600}$ units were collected for expression fractions (cells). Cell pellets were resuspended in (2×) Tris-Glycine SDS Sample Buffer (Novex, Life Sciences) supplemented with 5% (vol/vol) β-mercaptoethanol. For secretion fraction (media), culture volumes equivalent to 10 $OD_{600}$ units were filtered (0.22 μm), and proteins were precipitated using the deoxycholate and trichloroacetic acid method [76]. Protein pellets were washed twice with cold acetone, air-dried for 10 min, and then resuspended in 20 μL of 10 mM Tris-HCl pH = 8.0, followed by the addition of 20 μL of (2×) Tris-Glycine SDS Sample Buffer supplemented with 5% (vol/vol) β-mercaptoethanol. Next, samples from the cells and media fractions were incubated at 95 °C for 10 or 5 min, respectively, and loaded onto TGX stain-free gels (Bio-Rad) for SDS-PAGE. Resolved proteins were

transferred onto 0.2 μm nitrocellulose membranes using Trans-Blot Turbo Transfer (Bio-Rad). Membranes were probed with custom-made anti-Hcp antibodies (GenScript; polyclonal antibodies raised in rabbits against the peptide DPQSGQPAGQRVHKC), anti-c-Myc antibodies (Santa Cruz Biotechnology, sc-40, 9E10), anti-FLAG antibodies (Sigma-Aldrich, F1804), or anti-GST antibodies (Santa Cruz Biotechnology; sc-459) at a 1:1,000 dilution. As a control for loading and lysis, membranes were also probed with Direct-Blot HRP anti-*E. coli* RNA Polymerase β Antibody (Bio-legend, clone 8RB13; referred to as α-RpoB) at a 1:40,000 dilution. Protein signals were visualized in a Fusion FX6 imaging system (Vilber Lourmat) using ECL.

## Comparative proteomics analyses

Supernatant samples were obtained as described in the "*Aeromonas* Secretion Assays" section from three biological replicates for each strain. Pelleted proteins from the supernatant fractions were kept in acetone and sent to the Smoler Proteomics Center at the Technion for subsequent mass spectrometry analyses.

**Proteolysis and mass spectrometry analysis.** The samples were precipitated in 80% acetone overnight and washed three times with 80% acetone. The protein pellets were dissolved in 8.5 M urea and 400 mM ammonium bicarbonate. Protein concentrations were estimated using Bradford readings. The proteins were reduced with 10 mM DTT (60 °C for 30 min), modified with 40 mM iodoacetamide in 100 mM ammonium bicarbonate (at room temperature for 30 min in the dark), and digested overnight at 37 °C in 1.5 M urea and 66 mM ammonium bicarbonate with modified trypsin (Promega) in a 1:50 (M/M) enzyme-to-substrate ratio. An additional trypsin digestion step was performed for 4 h at 37 °C in a 1:100 (M/M) enzyme-to-substrate ratio. The tryptic peptides were desalted using C18 tips (homemade stage tips), dried, and re-suspended in 2% ACN\H$_2$O\0.1% Formic acid. The resulting peptides were analyzed by LC-MS/MS using a Q Exactive HF mass spectrometer (Thermo) fitted with a capillary HPLC (Evosep). The peptides were loaded onto a 15 cm, ID 150 μm, 1.9-micrometer (Batch no. E1121-3-24) column of Evosep. The peptides were eluted with the built-in Xcalibur 15 SPD (88 min) method. Mass spectrometry was performed in a positive mode using repetitively full MS scan (*m/z* 300–1,500) followed by High energy Collision Dissociation of the 20 most dominant ions selected from the full MS scan. A dynamic exclusion list was enabled with an exclusion duration of 20 s.

**Mass spectrometry data analysis.** The MaxQuant software 2.1.1.0 [77] was used for peak picking and identification using the Andromeda search engine, searching against a locally annotated genome of *A. jandaei* DSM 7311, available in S4 File, with mass tolerance of 6 ppm for the precursor masses and 20 ppm for the fragment ions. Oxidation on methionine and protein N-terminus acetylation were accepted as variable modifications and carbamidomethyl on cysteine was accepted as static modifications. The minimal peptide length was set to 7 amino acids and a maximum of two miscleavages was allowed. Peptide- and protein-level false discovery rates were filtered to 1% using the target-decoy strategy. The data was quantified by label-free analysis. Protein tables were filtered to eliminate the identifications from the reverse database and common contaminants. Statistical analysis of the identification and quantization results was done using the Perseus 1.6.7.0 software [78] (Mathias Mann's group).

## Bacterial competition assays

Bacterial competitions were performed as described previously [79]. Briefly, attacker and prey strains were grown overnight in appropriate media. Bacterial cultures were normalized to an OD$_{600}$ = 0.5, and then mixed at a 4:1 (attacker:prey) ratio in triplicate. The mixtures were spotted on LB agar plates and incubated for 4 h at 30 °C. The plates were supplemented with

0.05%–0.1% (wt/vol) L-arabinose as indicated when expression from an arabinose-inducible plasmid was required. The viability of the prey strain was determined as colony-forming units growing on selective plates at the 0- and 4-h time points.

### Toxicity assays in *E. coli*

To examine the toxic effects of Awe1 and its derivatives, *E. coli* MG1655 strains carrying arabinose-inducible expression plasmids encoding the indicated proteins were grown overnight in 2xYT media supplemented with kanamycin (30 μg/mL) and 0.04% (wt/vol) D-glucose (to repress leaky expression from the *Pbad* promoter). Bacterial cultures were washed twice (to remove residual glucose) and normalized to $OD_{600}$ = 0.01 in 2xYT media supplemented with kanamycin (30 μg/mL). From each sample, 200 μL were transferred into 96-well plates in quadruplicate. Cultures were grown at 30 °C in a BioTek SYNERGY H1 microplate reader with constant shaking (205 cpm). After 2 h, L-arabinose was added to each well to a final concentration of 0.05% (wt/vol), to induce expression from the plasmids. $OD_{600}$ readings were taken every 10 min; the data were visualized using GraphPad PRISM.

### Protein expression *in E. coli*

To examine the toxic effects of Awe1 and its derivatives, *E. coli* MG1655 strains carrying arabinose-inducible expression plasmids encoding the indicated proteins were grown overnight in 2xYT media supplemented with kanamycin (30 μg/mL) and 0.04% (wt/vol) D-glucose (to repress leaky expression from the *Pbad* promoter). Bacterial cultures were washed twice (to remove residual glucose) and normalized to $OD_{600}$ = 0.5 in 3 mL 2xYT media containing kanamycin (30 μg/mL). Bacterial cultures were incubated with agitation at 30 °C for 1.5 h, and then L-arabinose was added to a final concentration of 0.05% (wt/vol) to induce protein expression. Bacterial cultures were grown for 1.5 additional hours, and then 0.5 $OD_{600}$ units were collected, and pellets were resuspended in (2×) Tris-Glycine SDS Sample Buffer (Novex, Life Sciences) supplemented with 5% (vol/vol) β-mercaptoethanol. Next, samples were incubated at 95 °C for 10 min and loaded onto TGX stain-free gels (Bio-Rad) for SDS-PAGE. Transfer onto 0.2 μm nitrocellulose membranes was performed using Trans-Blot Turbo Transfer (Bio-Rad). Membranes were probed with anti-c-Myc antibodies (Santa Cruz Biotechnology, sc-40, 9E10) at a 1:1,000 dilution. Protein signals were visualized in a Fusion FX6 imaging system (Vilber Lourmat) using ECL.

### Co-immunoprecipitation assays

To investigate the interaction between the Awe1 WHIX domain and VgrG4, *E. coli* BL21 (DE3) strains containing plasmids for the expression of N-terminal FLAG-tagged VgrG4 or its truncated form (VgrG4[1–680]), and C-terminal Myc-tagged Awe1 WHIX domain (Awe1[149–699]) or superfolder GFP were grown overnight in 2xYT media supplemented with kanamycin (30 μg/mL) and chloramphenicol (10 μg/mL) at 37 °C. The overnight cultures were diluted 1:100 into fresh media and incubated at 37 °C for 2 h. Then, 0.1% (wt/vol) L-arabinose was added to induce protein expression, and the cultures were further incubated at 30 °C for 4 h. Next, 100 $OD_{600}$ units of the cultures were harvested by centrifugation at 4,000 *g* for 10 min at 4 °C. The cell pellets were resuspended in 3 mL of lysis buffer (500 mM NaCl, 20 mM Tris-HCl pH 7.5, 1 mM EDTA, and 0.5% NP-40 [vol/vol]) containing 0.1 mM PMSF, and the cells were lysed using a high-pressure homogenizer (Multi-cycle cell disruptor, Constant Systems). The resulting cell debris was removed by centrifugation at 15,000 *g* for 20 min at 4 °C. For the input fraction, 50 μL of the supernatant was mixed with 50 μL of (2×) protein sample buffer and kept for subsequent analysis. For immunoprecipitation, 495 μL of the supernatant were

incubated with 5 μL of anti-c-Myc antibody (Santa Cruz Biotechnology, sc-40, 9E10) for 1 h at 4 °C with continuous rotation. Protein G magnetic beads (NEB, S1430S) were prewashed with lysis buffer, and 12.5 μL were added to the supernatant and antibody samples and incubated for 3 h at 4 °C with continuous rotation. The beads were washed 10 times with 800 μL of lysis buffer and 1-minute rotations at room temperature. For the output fraction, the beads were collected, and the bound proteins were released by adding 100 μL of (2×) Tris-Glycine SDS sample buffer supplemented with 5% (vol/vol) β-mercaptoethanol, followed by heating at 70 °C for 5 min. The samples were then analyzed by immunoblotting, as described in "*Aeromonas* Secretion Assays". HRP-conjugated α-Light Chain-specific secondary antibodies (Jackson ImmunoResearch) were used to prevent the detection of the primary antibodies' heavy chain.

### Bacterial two-hybrid assays

To investigate the interaction between the Awe1 N- and C-terminal domains and the predicted immunity proteins, plasmids encoding the indicated proteins fused to either the T25 or T18 domains of the *Bordetella* adenylate cyclase were constructed. The resulting plasmids were co-transformed into the *E. coli* BTH101 reporter strain. Resulting colonies were grown overnight in a 96-well plate containing 2xYT media supplemented with kanamycin (30 μg/mL), ampicillin (100 μg/mL), and IPTG (0.5 mM) at 30 °C. The following day, 5 μL of each overnight culture were spotted on LB agar plates containing kanamycin (30 μg/mL), ampicillin (100 μg/mL), IPTG (0.5 mM), and X-gal (40 mg/mL). The plates were incubated for 24 h at 30 °C. The experiment was repeated three times with similar results.

### Identifying WHIX domain-containing proteins

PSI–BLAST was employed to construct the position-specific scoring matrix (PSSM) of WHIX. Five iterations of PSI–BLAST were performed against the RefSeq protein database using amino acids 1–477 of WP_005373349.1 (from *V. alginolyticus* 12G01). A maximum of 500 hits with an expect value threshold of $10^{-6}$ and 70% query coverage were used in each iteration.

A local database containing the RefSeq bacterial nucleotide and protein sequences was built (last updated on August 21, 2023). RPS–BLAST was used to identify WHIX domain-containing proteins in the local database. The results were filtered using an expect value threshold of $10^{-6}$ and a 70% overall coverage of the WHIX domain. In cases where more than one hit was identified in a protein accession, the boundaries of the WHIX domain in the protein accession were determined based on the starting position of the first hit and the ending position of the last hit.

Analysis of the genomic neighborhoods of WHIX domain-containing proteins (S1 Dataset) was performed as described previously [80,81]. Unique protein accessions located at the ends of the contigs were removed. To avoid duplications, protein accessions appearing in the same genome in more than one genomic accession were removed if the same downstream protein existed at the same distance from the protein. Identification of the T6SS core components [6] in WHIX domain-containing genomes (S2 Dataset) was performed as previously described [32]. Genomes encoding at least 9 out of the 11 T6SS core components were regarded as harboring T6SS (T6SS⁺). Protein accessions were analyzed using the NCBI Conserved Domain Database [64], SignalP v5.0 [82], and Phobius v1.01 [61].

### Illustration of conserved residues in WHIX domains

WHIX domain sequences were aligned using Clustal-Omega v1.2.4 [83]. Aligned columns not found in the WHIX domain of WP_005373349.1, the protein accession that was used to generate the PSSM of WHIX, were discarded. The conserved WHIX domain residues were illustrated using the WebLogo 3 server [84] (https://weblogo.threeplusone.com).

### Constructing a phylogenetic tree of WHIX domain-containing proteins

Q-TREE v2.2.2.6 [85] was employed to construct a Maximum-Likelihood phylogenetic tree of WHIX domain-containing proteins. The multiple sequence alignment of the WHIX domain, generated using Clustal-Omega, was used as input. 4,433 sequences with 2,484 amino acid sites were analyzed (360 invariant sites, 1,881 parsimony informative sites, and 2,305 distinct site patterns). ModelFinder [86] was used to test various protein models. The best-fit model, JTT + I + G4, was chosen according to the Bayesian Information Criterion. The tree was visualized using iTOL [87] (https://itol.embl.de/).

### Analyses of domains fused to WHIX

To identify N-terminal and C-terminal extension domains in WHIX effectors, we clustered the N-terminal and C-terminal (last 100 amino acids) sequences separately in two dimensions using the CLANS application [88,89]. Activities or domains in each cluster were identified by analyzing at least two representative sequences from each cluster using the NCBI CDD [64], followed by analysis using HHpred [88] if no domain was apparent in CDD.

### Protein structure predictions

The structure prediction of the *V. alginolyticus* effector WP_005373349.1, Awe1 (WP_082035413.1), AwiU (WP_042029887.1; amino acids 22–132), and AwiD (WP_156128651.1; amino acids 17–120) were carried out using AlphaFold 3 [53] (https://alphafoldserver.com/). The best model was selected and visualized using ChimeraX [90] version 1.7.1 together with its Predicted Aligned Error (PAE) plot.

The structure prediction of the complex formed by monomeric Awe1 and trimeric VgrG4 was carried out using the AlphaFold-Multimer 2 [91,92] with AlphaFold Protein Structure Database (released on July 2022). The prediction process involved generating multiple models, with the best model selected based on the lowest PAE values. The final model and PAE plots were visualized using ChimeraX version 1.5. The specific interactions between Awe1 and VgrG4 were further analyzed by examining the predicted interfaces. Residues of Awe1 and VgrG predicted to interact were identified and highlighted in green and orange, respectively. All computations were performed on an in-house computer with an NVIDIA GeForce RTX 3080 GPU and 250 GB of RAM.

### Whole genome sequencing

*A. jandaei* DSM 7311 was purchased from DSMZ (https://www.dsmz.de/). Genomic DNA was isolated using the Presto mini gDNA Bacteria kit (Geneaid), and DNA was sent to Plasmidsaurus (https://www.plasmidsaurus.com/) for their bacterial genome sequencing service using Oxford Nanopore Technologies long-read sequencing technology. Briefly, the process included: (i) construction of an amplification-free long read sequencing library using v14 library prep chemistry, (ii) sequencing the library using R10.4.1 flow cells, (iii) assembling the genome using the Plasmidsaurus standard pipeline, and (iv) genome annotation using Bakta [93].

The genome sequence is available as NCBI RefSeq assembly GCF_037890695.1; Chromosome RefSeq accession NZ_CP149571.1.

### Statistical analyses

All the statistical analyses in bacterial competition assays were performed with log-transformed data. Each data point is presented, and the mean ± standard deviation is shown in the figures. Additional details on statistical significance and tests are specified in the figure legends.

## Supporting information

**S1 Table. Bacterial strains used in this study.**
(DOCX)

**S2 Table. Plasmids used in this study.**
(DOCX)

**S1 Fig. C-terminal and N-terminal domains fused to WHIX comprise diverse peptidoglycan-targeting enzymes.** Sequences C-terminal (last 100 amino acids) **(A)** or N-terminal **(B)** to WHIX domains were clustered in two dimensions based on all-against-all sequence similarity, using the CLANS application, with nodes representing unique sequences and connecting lines representing the distances between sequences. The predicted activities or domains identified in each cluster are denoted and color-coded according to the nodes. In (A), black nodes (others) comprise clusters of <4 members, which were not analyzed. The data underlying this figure can be found in S2 File (A) and S3 File (B).
(TIF)

**S2 Fig. *Aeromonas* jandaei DSM 7311 has an antibacterial T6SS that secretes four effectors. (A)** Schematic representation of *A. jandaei* DSM 7311 T6SS gene cluster and auxiliary operons. Predicted protein activity is denoted above. **(B and C)** Expression (cells) and secretion (media) of Hcp from wild-type (WT; T6SS$^+$) *A. jandaei* DSM 7311 and a T6SS$^-$ mutant strain (Δ*tssB*) grown for 3 h at the indicated temperatures in media containing 1% or 3% (wt/vol) NaCl (LB or MLB, respectively). RNA polymerase beta subunit (RpoB) was used as a loading and lysis control. Results from a representative experiment out of at least three independent experiments are shown. In (C), bacterial strains contain either an empty plasmid (pEmpty) or a plasmid for the arabinose-inducible expression of *tssB* (pTssB), and the assay was performed at 30 °C in LB media supplemented with chloramphenicol and 0.1% (wt/vol) L-arabinose. **(D)** Viability counts (colony forming units; CFU) of *E. coli* MG1655 prey strains before (0 h) and after (4 h) co-incubation with the indicated *A. jandaei* DSM 7311 attacker strains containing an empty plasmid (pEmpty) or a plasmid for the arabinose-inducible expression of *tssB* (pTssB), on LB plates supplemented with 0.1% (wt/vol) L-arabinose at 30 °C. The statistical significance between samples was calculated using two-way ANOVA with Tukey multiple comparisons test on log-transformed data; ****$P < 0.0001$; *$P = 0.0353$ or 0.0362; ns, no statistical significance ($P > 0.05$); WT, wild-type; DL, the assay's detection limit. Data are shown as the mean ± SD; $n = 3$. The data shown are a representative experiment out of at least three independent experiments. The data underlying this figure can be found in S4 Data. **(E)** Volcano plot summarizing the comparative analysis of proteins identified in the media of *A. jandaei* DSM 7311 WT and T6SS$^-$ (Δ*tssB*) strains, using label-free quantification (LFQ). The average LFQ signal intensity difference between the WT and T6SS$^-$ strains is plotted against the $-\text{Log}_{10}$ of Student *t*-test *P*-values ($n = 3$ biological replicates). Proteins that were significantly more abundant in the secretome of the WT strain (difference in average LFQ intensities > 3; *P*-value < 0.02; with a score > 40) are denoted in red.
(TIF)

**S3 Fig. TseI, Tle1, and DUF3289 are antibacterial T6SS effectors.** Viability counts (CFU) of *A. jandaei DSM 7311 prey strains in which the genes encoding TseI (A), Tle1 (B), and DUF3289 (C) were deleted with their predicted downstream immunity genes. Prey strains contain an empty plasmid (pEmpty) or a plasmid for the arabinose-inducible expression of the predicted cognate immunity protein (pImm), before (0 h) and after (4 h) co-incubation with the indicated *A. jandaei* DSM 7311 attacker strains on LB plates supplemented with 0.05% (wt/vol) L-arabinose at 30 °C. The statistical significance between samples at the 4-h

time point was calculated using an unpaired, two-tailed Student $t$-test on log-transformed data; ***$P < 0.0001$; **$P < 0.0002$; WT, wild-type; DL, the assay's detection limit. Data are shown as the mean ± SD; $n = 3$. The data shown are a representative experiment out of at least three independent experiments. The data underlying this figure can be found in S5 Data.
(TIF)

**S4 Fig. Awe1 N- and C-terminal domains are similar to peptidoglycan-degrading enzymes. (A)** Superimposed AlphaFold3 structure prediction of the N-terminal (N-ter) Awe1 domain (amino acids 1–147; colored orange) with *Vibrio* cholerae ShyA (PDB: 6UE4A; colored fiusha). **(B)** Superimposed AlphaFold3 structure prediction of the C-terminal (C-ter) Awe1 domain (amino acids 700–862; colored beige) with the AlphaFold structure prediction of *Vibrio parahaemolyticus* FlgJ (AlphaFold database [AFDB]: Q9X9J3; colored blue). The data underlying this figure can be found in S7 File.
(TIF)

**S5 Fig. Awe1 expression in *E. coli*.** The expression of the indicated C-terminally Myc-tagged superfolder GFP and Awe1 forms expressed in the cytoplasm (Cyto) or periplasm (Peri) of *Escherichia* coli MG1655 from an arabinose-inducible plasmid.
(TIF)

**S6 Fig. VgrG4 is not required for *Aeromonas* jandaei T6SS activity. (A)** Expression (cells) and secretion (media) of Hcp from wild-type (WT; T6SS$^+$) *A. jandaei* DSM 7311, a T6SS$^-$ mutant strain ($\Delta tssB$), and a *vgrG4* deletion strain ($\Delta vgrG4$) grown for 3 h at 30 °C in media containing 1% (wt/vol) NaCl (LB). RNA polymerase beta subunit (RpoB) was used as a loading and lysis control. Results from a representative experiment out of at least three independent experiments are shown. **(B)** Viability counts (colony forming units; CFU) of *E. coli* MG1655 prey strains before (0 h) and after (4 h) co-incubation with the indicated *A. jandaei* DSM 7311 attacker strains on LB plates at 30 °C. The statistical significance between samples at the 4-h time point was calculated using an unpaired, two-tailed Student $t$-test on log-transformed data; WT, wild-type; DL, the assay's detection limit. Data are shown as the mean ± SD; $n = 3$. The data shown are a representative experiment out of at least three independent experiments. The data underlying this figure can be found in S6 Data.
(TIF)

**S1 File. AlphaFold3 prediction of WP_005373349.1 structure.**
(ZIP)

**S2 File. CLANS analysis of sequences found C-terminal to WHIX.**
(ZIP)

**S3 File. CLANS analysis of sequences found N-terminal to WHIX in subclass II WHIX effectors.**
(ZIP)

**S4 File. Comparative proteomics mass spectrometry data analysis and the *Aeromonas jandaei* DSM 7311 genome annotation used for it.**
(ZIP)

**S5 File. AlphaFold3 prediction of Awe1 structure.**
(ZIP)

**S6 File. AlphaFold3 prediction of Awe1 structure in complex with AwiU and AwiD.**
(ZIP)

**S7 File. Superimposition of the Awe1 N- and C-terminal domains with peptidoglycan-degrading enzymes.**
(ZIP)

**S8 File. AlphaFold prediction of Awe1 structure in complex with a VgrG4 trimer.**
(ZIP)

**S1 Dataset. A list of WHIX-containing proteins and their genomic neighborhood.**
(XLSX)

**S2 Dataset. Identification of T6SSs in genomes encoding WHIX-containing proteins.**
(XLSX)

**S1 Data. Tree data for Fig 1.**
(ZIP)

**S2 Data. Numerical values for Fig 3.**
(XLSX)

**S3 Data. Numerical values for Fig 4.**
(XLSX)

**S4 Data. Numerical values for S2 Fig.**
(XLSX)

**S5 Data. Numerical values for S3 Fig.**
(XLSX)

**S6 Data. Numerical values for S6 Fig.**
(XLSX)

**S1 Raw Images. Uncropped and minimally adjusted images supporting all blot results included in the article.**
(PDF)

## Acknowledgments

We thank the Smoler Proteomics Center at the Technion for performing and analyzing the mass spectrometry data. This work was performed in partial fulfillment of the requirements for a PhD degree for Chaya Mushka Fridman at the Faculty of Medical and Health Sciences, Tel Aviv University.

## Author contributions

**Conceptualization:** Eran Bosis, Dor Salomon.

**Formal analysis:** Chaya Mushka Fridman, David Albesa-Jové, Eran Bosis, Dor Salomon.

**Funding acquisition:** David Albesa-Jové, Eran Bosis, Dor Salomon.

**Investigation:** Chaya Mushka Fridman, Kinga Keppel, Vladislav Rudenko, Jon Altuna-Alvarez, Eran Bosis, Dor Salomon.

**Methodology:** Chaya Mushka Fridman, David Albesa-Jové, Eran Bosis.

**Resources:** David Albesa-Jové, Eran Bosis.

**Supervision:** Eran Bosis, Dor Salomon.

**Writing – original draft:** Dor Salomon.

**Writing – review & editing:** Chaya Mushka Fridman, Kinga Keppel, Vladislav Rudenko, Jon Altuna-Alvarez, David Albesa-Jové, Eran Bosis.

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
