## [Editor Report · Decision Letter 0]

27 Aug 2024

Dear Dor,

I am happy to see another submission from your group. Thank you for submitting your manuscript entitled "WHIX is a T6SS secretion domain found in polymorphic double-edged sword effectors" for consideration as a Research Article by PLOS Biology.

Your manuscript has now been evaluated by the PLOS Biology editorial staff, as well as by an academic editor with relevant expertise, and I am writing to let you know that we would like to send your submission out for external peer review.

Once your full submission is complete, your paper will undergo a series of checks in preparation for peer review. After your manuscript has passed the checks it will be sent out for review. To provide the metadata for your submission, please Login to Editorial Manager (https://www.editorialmanager.com/pbiology) within two working days, i.e. by Aug 29 2024 11:59PM.

Kind regards,

Melissa

Melissa Vazquez Hernandez, Ph.D.

Associate Editor

PLOS Biology

---

## [Decision Letter · Decision Letter 1]

10 Oct 2024

Dear Dor,

Thank you for your patience while your manuscript "WHIX is a T6SS secretion domain found in polymorphic double-edged sword effectors" was peer-reviewed at PLOS Biology. It has now been evaluated by the PLOS Biology editors, an Academic Editor with relevant expertise, and by several independent reviewers. 

In light of the reviews, which you will find at the end of this email, we would like to invite you to revise the work to thoroughly address the reviewers' reports. As you will see below, majority of reviewers are positive about the relevance and novelty of the study, yet important concerns were raised during revision. Overall, most reviewers appreciate the relevance and novelty of your study, but key concerns remain. Reviewer #1 raised issues with the heavy reliance on computational predictions, particularly for the T6SS effectors secreted by A. jandaei. S/he suggests that the term "double-edged sword" may be misleading, as there is no evidence of self-harm. They also request additional controls, such as testing Hcp secretion or E. coli killing in the ∆VgrG4 strain, to confirm that VgrG4 does not disrupt T6SS activity. Reviewer #2 highlighted statistical concerns and the need for controls to support the claim that VgrG4 is necessary for Awe1 delivery. Reviewer #3 believes the evidence for a new family of effectors is insufficient and suggests deeper analysis of the WHIX protein toxic domains, quantification of immunoblots, and clarification of the mode of action and target of Awe1.

IMPORTANT: after discussion with the Academic Editor and the reviewers, we think that most requests from the reviewers should be addressed. However, as this is a Short Report, fully deciphering the function of Awe1 is not required. Please also revise the use of the term "double-edged sword" for clarity.

Given the extent of revision needed, we cannot make a decision about publication until we have seen the revised manuscript and your response to the reviewers' comments. Your revised manuscript is likely to be sent for further evaluation by all or a subset of the reviewers.

**IMPORTANT - SUBMITTING YOUR REVISION**

*Re-submission Checklist*

*Published Peer Review*

*PLOS Data Policy*

*Blot and Gel Data Policy*

Sincerely,

Melissa

Melissa Vazquez Hernandez, Ph.D.

Associate Editor

PLOS Biology

REVIEWERS' COMMENTS:

Reviewer #1: 

In this work, the authors have identified a new family of type VI secretion system effectors consisting of a VgrG-associating domain that can have additional functional domains fused to the N- or C-terminal of the domain. Although the results are interesting, there is a heavy reliance on computational predictions that are not fully experimentally confirmed. Additionally, the title using the phrase "double-edged sword" is very misleading.

Figure 2B - showing this figure is not particularly meaningful... T6SS seems to be active under all conditions tested. Considering there is variation amount of protein in the control between samples, it's not like one can make a definitive regarding when T6SS is "most" active (lines 177-179). 

The claim that "Aeromonas jandaei T6SS secretes four antibacterial effectors", is a bit weak. TseI, Tle1 and Awe1 are just hypothetical effectors based on "computational identification" and assumed activity. Only Awe1 and the 4th effector are actually demonstrated to be antibacterial effectors in this study. Also, how confident are the authors that there are not additional unidentified effectors?

Lines 280-281: The term "double-edged sword" generally refers to something that has both positive and negative effects (i.e. something that is both advantageous and disadvantageous). Using this term strongly implies that that WHIX are somehow self-harming... which is not really true. The WHIX domain, while literally having two "edges" is not really a "double-edged sword".

Figure 5 - how do you know that deleting VgrG4 doesn't disrupt T6SS activity? T6SS in other organisms have been demonstrated to have an essential VgrG, even when there are multiple VgrGs in play (e.g. V. cholerae). The control showing hcp secretion or even just E. coli killing in the ∆VgrG4 strain should be shown. Also, it would be nice to see the the WHIX construct (including the version with heterologous GST/GFP domains) being secreted with full-length VgrG4 but not with the truncated VgrG4.

Lines 360-362: VgrG4 being secreted in a T6SS-dependent manner (5C) does not imply anything about Awe1. This is "begging the question". The authors are presupposing that Awe1 must be interacting with VgrG4 without ever actually demonstrating the interaction and using that presupposition to conclude that Awe1 is associated with VgrG4.

Line 422-423: The authors hypothesize that carrying two toxic domains reduces the risk of resistance developing, however in their experiments, they explicitly demonstrate that only a single immunity protein AwiD is sufficient to confer protection against the entire polymorphic Awe1 effector under native conditions. Some discussion of this apparent contradiction is probably warranted.

Line 443-445: This speculation seems to come out of nowhere. What reason do the authors have to think that WHIX could play an active role?

I wasn't able to open Supporting File S2

Reviewer #2: 

The manuscript entitled "WHIX is a T6SS secretion domain found in polymorphic double-edged sword effectors" authored by Mushka et. al. details the identification of a domain which demonstrates the ability to carry two distinct effector proteins which function by targeting cell's periplasmic regions and peptidoglycan. While working with Aeromonas jandaei, the authors identified a T6SS effector protein (Awe1) which carries two toxic domains each with their own immunity proteins (AwiD and AwiU). The N-terminal toxic domain, though not effective against Aeromonas, was toxic to E. coli prey. A potential mechanism of secretion for WHIX-containing effector was suggested to be associated with the C-terminal tail of VgrG4. 

Overall, the manuscript is well written with a logical progression. Multiple methods were employed ranging from meta-sequence analysis, proteomics, protein structure, and more classic mutation-based experimentation with appropriate controls. However, the many weaknesses were noted and are summarized below.

Comments

1. Discovery of fourth effector protein (WP_042032936.1) with DUF3289 in the result section is interesting and probably should be part of the main manuscript [1-3] rather than the supplemental figures. It is also recommended to remove the results depicted in figure 5 and move supplemental figures S3 into the main body of the paper. Briefly mentioning these findings at the end of the results section in figure 5 leaves the reader disoriented from the main takeaways of the rest of the manuscript, lessens the impact of the other findings, and brings up more questions than it does answers.

2. Throughout the manuscript, the statistical significance test which was chosen to analyze the data was an unpaired, two-tailed, student T-test. This is not the appropriate test to use in any of the cases here, as multiple groups were being compared to each other (i.e., figure 5B, in which case a one-way ANOVA is called for) or in some cases, multiple variables across multiple groups (i.e., figure 4B in which case a two-way ANOVA is called for). Analyzing multiple groups using a student T-test, which is designed for comparisons between two groups alone, compounds the chances of making a type-1 error leading one to underestimate the error in the experiment. This can potentially lead to finding significance in data where there is none.

3. Line 356-357 draws the conclusion "these results indicate that VgrG4 is required for Awe1 delivery", which is an overarching claim for the experiment that was performed. Deleting vgrG4 may have just abolished the efficacy of the entire T6SS and therefore the prey was not intoxicated. This experiment needs to be repeated with a built-in control to prove the continued function of the T6SS after VgrG4 deletion to make the claim this protein is required for Awe1-specific secretion.

4. In the introduction, a background of the T6SS and its antibacterial use is given. However, no mention is made of the T6SS and its host-modulatory effects, a well-established phenomenon [4]. A mention of this in the introductory material would be useful, as in line 182 other roles besides "interbacterial competition" is inferred.

5. In figures 1A and 1E, the N-terminus and C-terminus should be visually indicated. This would help avoid any confusion potentially felt by the reader, especially one less familiar with standard amino acid sequence annotations.

6. In line 116, the 99 amino acid extension on the N-terminus of the subclass II WHIX-containing effectors is first mentioned. It is assumed this extension was the green triangle indicated in figure 1E as "immunity". It is suggested indicating this N-terminus region which divides subclass II WHIX-containing protein from subclass I with some sort of bracket and label in figure 1E to clearly indicate to the reader what is being referred to.

7. Lines 144-148 detail the T6SS gene cluster found in A. jandaei and mentions the existence of two 'auxiliary modules' each with its own effector protein. In figure 2A, a third module is depicted with a putative effector labeled "DUF3289". This effector is addressed later on in the manuscript, but at this point appeared to be left untouched. It is suggested adding a mention of it here, with further details later on. Same thing in line 180. Three effectors are mentioned but figure 2A shows four effectors, so this can be confusing.

8. Lines 230-231 describe the results of modeling the binding of immunity proteins AwiU and AwiD to Awe1. Does the prediction that the immunity proteins bind to the N- and C-terminal domains of the Awe1 effector have anything to do with the fact that, genomically they are located directly up and downstream of the awe1 gene? seems obvious that feeding the awe1 sequence along with the sequences of the immunity proteins into alphafold3, the program will predict the sequences on either end of awe1 be attached to those same ends after translation? Or is the input for this model three separate sequences and still the predicted binding in the same orientation they exist on the genome? It is not clear in the paper.

9. In the description for figure 3D in line 265, its mentioned that 'full-length Awe1' secretion was monitored. In reality, an anti-Myc-tagged version of full-length Awe1 secretion expressed from a plasmid was actually measured after the deletion of native awe1 in either T6SS+ or T6SS- strains. This wasn't clear until reading the figure legend and this detail should be mentioned in the body of the text.

10. The conclusion in line 344-345 should include a mention of the fact that the N-terminal domain's toxicity may be specifically directed at other species, for example E. coli.

11. A brief description of VgrG proteins in line 348 should be added as a reminder to readers, so it is clear the role these proteins play in the T6SS before getting into the results of this section.

12. In line 350, the AlphaFold structure of WHIX is cited as a justification for the hypothesis that Awe1 WHIX domain is loaded onto the C-terminal tail of VgrG4. The 'deep cavity' in this domain mentioned in earlier results is specifically pointed to in the discussion section but it needs to be mentioned here in the results section as to why WHIX may be binding to the C-terminal domain of VgrG4.

13. Are there any plans for future work to be done on the mechanism of action against peptidoglycan? It is not mentioned as a future direction in the discussion section, which looks like an oversight because the putative toxicity of these WHIX domains to peptidoglycan specifically is mentioned throughout the paper and peptidoglycan as a target is even given as a final conclusion of the whole paper at the end in lines 455-457.

14. The second to last paragraph in lines 446-452 is just a restatement of results already given in the results section. More needs to be said in this paragraph about the implications and these results, and speculations for future work it spurs, and its impact on the field.

References

1. Jana, B., et al., A modular effector with a DNase domain and a marker for T6SS substrates. Nat Commun, 2019. 10(1): p. 3595.

2. Fitzsimons, T.C., et al., Identification of Novel Acinetobacter baumannii Type VI Secretion System Antibacterial Effector and Immunity Pairs. Infect Immun, 2018. 86(8).

3. Li, D.Y., et al., Identification and Characterization of EvpQ, a Novel T6SS Effector Encoded on a Mobile Genetic Element in. Front Microbiol, 2021. 12: p. 643498.

4. Singh, R.P. and K. Kumari, Bacterial type VI secretion system (T6SS): an evolved molecular weapon with diverse functionality. Biotechnol Lett, 2023. 45(3): p. 309-331.

Reviewer #3: 

Fridman and coworkers aim to shed light into a new domain, WHIX, found in T6SS effectors. Effectors containing this domain may have one or two toxic domains, and there seems to be distinct immunity proteins. Authors postulate that WHIX acts as a cargo domain. Additionally, authors provide an initial characterization of Aermonas jandaei T6SS system. 

General comment.

It adds to the T6SS field the characterization of a putative new cargo domain although the evidence is not solid enough yet to make the point that a new family of effectors has been identified. What authors have described is common to many effectors across (toxic domains with immunity protiens around and/or closely associated with a VgrG protein). No characterization of the target and mode of action of the effector is presented and much of the structural work relies on in silico predictions with limited experimental validation. The section on Aermonas general characterization of the T6SS while well done distracts from the message of the paper and in fact does not add much to the overall notion of the work. Hope that the comments will help the authors to strengthen the case they are trying to make. 

Major comments.

1. To further strengthen the analysis in Fig 1B, authors should analyze the distribution of the the toxic domains in the putative WHIX protein identified: (i) which is the prevalence of one and two putative toxic domains? which is the diversity of toxic domains? is there a topic of toxic domain over represented?

2. Fig 2B. To rigorously make the case, authors would need to quantify the 3 immunoblots, and add data related to differences of transcription between temperature and NaCl content. 

3. Authors may need to revise the statistical analysis of the manuscript. In most cases (Fig 2, 4 and 5) it is more appropriate a two-side ANPOVA with multiple comparisons correction. 

4. The binding of the immunity proteins to Awe1 can be tested experimentally by, for example, bacteria two-hybrid experiments. Additionally, authors need to establish whether each of the immunity proteins bind to the cognate toxic domain. 

5. Fig 3D. Authors need to quantify the 3 immunoblots. The results that there is a reduction in the levels of secreted effector in all the truncated forms, particularly noticeable in the one lacking the N-terminal domain. If this is so, this will impact the killing experiments presented later on in the manuscript. 

6. Which is the Hcp responsible for the secretion of Awe1? Is there any promiscuity in terms of the VgrG that can secrete Awe1? It will be crucial to demonstrate whether indeed WHIX domain interacts with VgrG and the specificity of interaction with the 14 C-terminal AAs. As noted before, bacteria two hyvrid epxerimets may be useful here coupled with secretion experiments.

7. Fig 3E, as noted before authors need to quantify the 3 immunoblots. These constructs will also help authors to clarify the previous comment on secretion of the effector. Here it seems that the GST-WHIX-GFP chimeric protein is the best secreted. 

8. Authors need to establish the target and mode of action of Awe1, and assess whether each of the domains have a different function or they complement each other. This is essential to sustain the claim of a double sword effector. If both domains have the same target and mode of action then it will be challenging to justify the claim of the authors.

Minor comments.

1. In Figure 1, a pictorial representation of the class I and II would help to better appreciate the differences.

2. Based on the proteomic analysis, which is teh connection between T6SS and motility iin Aeromonas? Is a T6SS mutant motile?

Reviewer #4: 

In this manuscript, the authors identify a conserved domain (named WHIX) that is present in thousands of proteins encoded on genomes of T6SS positive bacteria. The authors show that this conserved domain can be extended on both the C- and N-terminus with additional protein domains that often represent putative effectors. In addition, periplasm targeted immunity proteins are often encoded next to such effectors. The authors then identify an active T6SS cluster in Aeromonas jandaei, which secretes an effector (Awe1) with this new domain in a T6SS dependent manner and can kill E. coli. Interestingly, this effector has both N- and C-terminal domains that are potentially toxic to target cells. The authors showed that those two domains can be removed without affecting secretion and even replaced with GST and GFP, which are then secreted in a T6SS-dependent manner. The effector as well as the two domains are toxic when expressed in the periplasm of E. coli but only one has a major impact on the outcome of competition assays. Finally, this novel effector requires a cognate VgrG (its last 14 aa) to get secreted and delivered into target cells. Overall, this manuscript is clearly written, the data is solid and well presented, there are appropriate controls available and thus the conclusions are justified. It is an interesting and important addition to our understanding of T6SS effector reservoir. I have no major comments and I congratulate the authors on their manuscript.

---

## [Editor Report · Decision Letter 2]

3 Feb 2025

Dear Dor,

Thank you for your patience while we considered your revised manuscript "WHIX is a T6SS secretion domain found in polymorphic double-blade effectors" for publication as a Short Reports at PLOS Biology. This revised version of your manuscript has been evaluated by the PLOS Biology editors, and the Academic Editor.

Based on our Academic Editor's assessment of your revision, we are likely to accept this manuscript for publication, provided you satisfactorily address the remaining editorial points. Please also make sure to address the following data and other policy-related requests.

a) We routinely suggest changes to titles to ensure maximum accessibility for a broad, non-specialist readership, and to ensure they reflect the contents of the paper. In this case, we would suggest a minor edit to the title, as follows. Please ensure you change both the manuscript file and the online submission system, as they need to match for final acceptance:

"A new class of type 6 secretion system effectors can carry two toxic domains and are recognized through the WHIX motif for export"

b) The maximum number of figures for a Short Report is 4. Currently you have 5. Could you please reduce it to 5 either by sending one figure to the supplements or by combining with the others? Thank you!

c) Thank you so much for sharing all the numerical values for the figures. Only the values for Fig 4A is missing, could you please provide it?

d) Thank you also for sharing all the raw images from the gels. I would only like to point out that the images for 3EF might be mislabeled

e) Please ensure that your Data Statement in the submission system accurately describes where your data can be found and is in final format, as it will be published as written there.

f) Per journal policy, if you have generated any custom code during the course of this investigation, please make it available without restrictions upon publication. Please ensure that the code is sufficiently well documented and reusable, and that your Data Statement in the Editorial Manager submission system accurately describes where your code can be found.

We expect to receive your revised manuscript within two weeks. 

*Published Peer Review History*

*Press*

Sincerely,

Melissa

Melissa Vazquez Hernandez, Ph.D.

Associate Editor

PLOS Biology

---

## [Editor Report · Decision Letter 3]

5 Feb 2025

Dear Dor,

Thank you for the submission of your revised Short Reports "A new class of type 6 secretion system effectors can carry two toxic domains and are recognized through the WHIX motif for export" for publication in PLOS Biology. On behalf of my colleagues and the Academic Editor, Sebastian Winter, I am pleased to say that we can in principle accept your manuscript for publication, provided you address any remaining formatting and reporting issues. These will be detailed in an email you should receive within 2-3 business days from our colleagues in the journal operations team; no action is required from you until then. Please note that we will not be able to formally accept your manuscript and schedule it for publication until you have completed any requested changes.

PRESS

Sincerely, 

Melissa

Melissa Vazquez Hernandez, Ph.D., Ph.D.

Associate Editor

PLOS Biology
